# Partial Transportability for Domain Generalization

**Kasra Jalaldoust**\*      **Alexis Bellot**\*†      **Elias Bareinboim**

Causal Artificial Intelligence Lab

Columbia University

{kasra, eb}@cs.columbia.edu, abellot95@gmail.com

## Abstract

A fundamental task in AI is providing performance guarantees for predictions made in unseen domains. In practice, there can be substantial uncertainty about the distribution of new data, and corresponding variability in the performance of existing predictors. Building on the theory of partial identification and transportability, this paper introduces new results for bounding the value of a functional of the target distribution, such as the generalization error of a classifier, given data from source domains and assumptions about the data generating mechanisms, encoded in causal diagrams. Our contribution is to provide the first general estimation technique for transportability problems, adapting existing parameterization schemes such Neural Causal Models to encode the structural constraints necessary for cross-population inference. We demonstrate the expressiveness and consistency of this procedure and further propose a gradient-based optimization scheme for making scalable inferences in practice. Our results are corroborated with experiments.

## 1 Introduction

In the empirical sciences, the value of scientific theories arguably depends on their ability to make predictions in a domain different from where the theory was initially learned. Understanding when and how a conclusion in one domain, such as a statistical association, can be generalized to a novel, unseen domain has taken a fundamental role in the philosophy of biological and social sciences in the early 21st century. As Campbell and Stanley [8, p. 17] observed in an early discussion on the interpretation of statistical inferences, "Generalization always turns out to involve generalization into a realm not represented in one's sample" where, in particular, statistical associations and distributions might differ, presenting a fundamental challenge.

As society transitions to become more AI centric, many of the every-day tasks based on predictions are increasingly delegated to automated systems. Such developments make various parts of society more efficient, but also require a notion of performance guarantee that is critical for the safety of AI, in which the problem of generalization appears under different forms. For instance, one critical task in the field is domain generalization, where one tries to learn a model (e.g. classifier, regressor) on data sampled from a distribution that differs in several aspects from that expected when deploying the model in practice. In this context, generalization guarantees must build on knowledge or assumptions on the "relatedness" of different training and testing domains; for instance, if training and testing domains are arbitrarily different, no generalization guarantees can be expected from any predictor [12, 40]. The question becomes how to link the domains of data that are used to train a model (a.k.a., the source domains) to the domain where this model is deployed in practice (a.k.a., the target domain).

---

\*Equal Contribution.

†Now at Google DeepMind.

38th Conference on Neural Information Processing Systems (NeurIPS 2024).

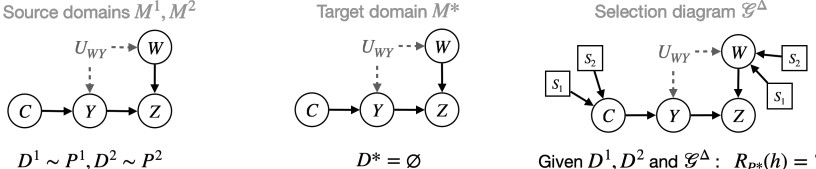

Figure 1: Illustration of the task of evaluating the generalization error of a model $h$. The mechanisms for $C$ and $W$ vary across domains.

To begin to answer this question, a popular type of assumption that relates source and target domains is *statistical* in nature: invariances in the marginal or conditional distribution of some variables across the source and target distributions. Examples include assumptions of covariate shift and label shift (among others) [35, 34]. Notably, generalization is justified by the stability and invariance of the causal mechanisms shared across the domains [14, 21], since the distributional/statistical invariances across the domains are consequences of mechanistic/structural invariances governing the underlying data generating process. Although the induced statistical invariances, once exploited correctly, can be used as bases for generalizability. Broadly, invariance-based approaches to domain generalization [27, 29, 2, 40, 24, 20, 7, 6, 13] search for predictors that not only achieves small error on the source data but also maintain certain notions of distributional invariance across the source domains. Since these statistical invariances can be viewed as proxies to structural invariances, in certain instances generalization guarantees can be provided through causal reasoning [17, 31, 39]. This idea can be illustrated in Fig. 1. The value of variables $\{C, Y, W, Z\}$ are determined as a stochastic function of variables pointing to it, while these functions may differ across domains. The challenge is to evaluate the generalization risk of a model, e.g. $R_{P*}(h) := \mathbb{E}_{P*}[(Y - h)^2]$ for $h := h(C, W, Z) = \mathbb{E}_{P^1}[Y \mid C, W, Z]$, without observations from the target $P*$. General instances of this challenge have been studied under the rubric of the theory of causal transportability, where qualitative assumptions regarding the underlying structural causal models are encoded in a graphical object, and algorithms are designed to leverage these assumptions and compute certain statistical queries in the target domain in terms of the existing source data [26, 4, 5, 19, 11, 17].

Despite these advances, in practice, the combination of source data and graphical assumptions is not always sufficient to identify (uniquely evaluate) the desired statistical query, e.g., the average loss of a given predictor in the target domain. In this case, the query is said to be non-transportable[3]. For example, given Fig. 1, $R_{P*}(h)$ is non-transportable for the classifier $h := h(C, W, Z)$. In this paper, we study the fundamental task of computing tight upper-bounds for statistical queries in a new unseen domain. This allows us to assess worst-case performance of prediction models for the domain generalization task. Our contributions are as follows:

- **Sections 2 & 3.** We develop the first general estimation technique for bounding the value of queries across multiple domains (e.g., the generalization risk) in non-transportable settings (Def. 4). Specifically, we extend the formulation of canonical models [3, 42] to encode the constraints necessary for solving the transportability task, and demonstrate their expressiveness for generating distributions entailed by the underlying Structural Causal Models (SCMs) (Thm. 1).

- **Section 4.** We adapt Neural Causal Models (NCMs) [41] for the transportability task via a parameter sharing scheme (Thm. 2), similarly demonstrating their expressiveness and consistency for solving the partial transportability task. We then leverage the theoretical findings in sections 2 & 3 to implement a gradient-based optimization algorithm for making scalable inferences (Alg. 1), as well as a Bayesian inference procedure. Finally, we introduce Causal Robust Optimization (CRO) (Alg. 2), an iterative method to find a predictor with the best worst-case risk.

**Preliminaries.** We use capital letters to denote variables ($X$), small letters for their values ($x$), bold letters for sets of variables ($\boldsymbol{X}$) and their values ($\boldsymbol{x}$), and use $\mathrm{supp}$ to denote their domains of definition ($x \in \mathrm{supp}_X$). A conditional independence statement in distribution $P$ is written as $(\boldsymbol{X} \perp\!\!\!\perp \boldsymbol{Y} \mid \boldsymbol{Z})_P$. A $d$-separation statement in some graph $\mathcal{G}$ is written as $(\boldsymbol{X} \perp\!\!\!\perp_d \boldsymbol{Y} \mid \boldsymbol{Z})$. To denote $P(\boldsymbol{Y} = \boldsymbol{y} \mid \boldsymbol{X} = \boldsymbol{x})$, we use the shorthand $P(\boldsymbol{y} \mid \boldsymbol{x})$. The basic semantic framework of our analysis relies on Structural Causal Models (SCMs) [25, Definition 7.1.1], which are defined below.

---

[3]The notion of non-transportability formalizes a type of aleatoric uncertainty [16] arising from the inherent variability within compatible data generating systems for the target domain. In particular, it cannot be explained away with increasing sample size from the source domains.

**Definition 1.** *An SCM $M$ is a tuple $M = \langle \boldsymbol{V}, \boldsymbol{U}, \mathcal{F}, P \rangle$ where each observed variable $V \in \boldsymbol{V}$ is a deterministic function of a subset of variables $\boldsymbol{Pa}_V \subset \boldsymbol{V}$ and latent variables $\boldsymbol{U}_V \subset \boldsymbol{U}$, i.e., $v := f_V(\boldsymbol{pa}_V, \boldsymbol{u}_V), f_V \in \mathcal{F}$. Each latent variable $U \in \boldsymbol{U}$ is distributed according to a probability measure $P(u)$. We assume the model to be recursive, i.e. that there are no cyclic dependencies among the variables.* □

SCM $M$ entails a probability distribution $P^{\mathcal{M}}(\boldsymbol{v})$ over the set of observed variables $\boldsymbol{V}$ such that

$$P^{\mathcal{M}}(\boldsymbol{v}) = \int_{\text{supp}_{\boldsymbol{U}}} \prod_{V \in \boldsymbol{V}} P^{\mathcal{M}}(v \mid pa_V, \boldsymbol{u}_V) \cdot P(\boldsymbol{u}) \cdot d\boldsymbol{u}, \tag{1}$$

where the term $P(v \mid pa_V, \boldsymbol{u}_V)$ corresponds to the function $f_V \in \mathcal{F}$ in the underlying structural causal model $M$. It also induces a causal diagram $\mathcal{G}_{\mathcal{M}}$ in which each $V \in \boldsymbol{V}$ is associated with a vertex, and we draw a directed edge between two variables $V_i \rightarrow V_j$ if $V_i$ appears as an argument of $f_{V_j}$ in the SCM, and a bi-directed edge $V_i \leftrightarrow V_j$ if $\boldsymbol{U}_{V_i} \cap \boldsymbol{U}_{V_j} \neq \emptyset$, that is $V_i$ and $V_j$ share an unobserved confounder. Throughout this paper, we assume the observational distributions entailed by the SCMs satisfy the positivity assumption, that is, $P^M(\boldsymbol{v}) > 0$, for every $\boldsymbol{v}$. We will also operate non-parametrically, i.e., making no assumption about the particular functional form or the distribution of the unobserved variables.

## 2 Risk Evaluation through Partial Transportability

In this section, we focus on challenges of the domain generalization problem through a causal lens, in particular regarding assessment of average loss of a given classifier in the target domain. We study a system of variables $\boldsymbol{V}$ where $Y \in \boldsymbol{V}$ is a categorical outcome variable and consider a classifier $h : \text{supp}_{\boldsymbol{X}} \rightarrow \text{supp}_Y$ mapping a set of covariates $\boldsymbol{X} \subset \boldsymbol{V}$ to the domain of the outcome. The setting of domain generalization is characterized by multiple domains, defined by SCMs $\mathbb{M} : \{\mathcal{M}^1, \ldots, \mathcal{M}^K, \mathcal{M}^*\}$ that entail distributions $\mathbb{P} = \{P^{\mathcal{M}^1}, \ldots, P^{\mathcal{M}^K}\}$ and $P^{\mathcal{M}^*}$ over $\boldsymbol{V}$. We are given a classifier $h : \text{supp}_{\boldsymbol{X}} \rightarrow \text{supp}_Y$, and the objective is to evaluate its risk in the domain $\mathcal{M}^*$ which is defined as,

$$R_{P*}(h) := \mathbb{E}_{P*}[\mathcal{L}(Y, h(\boldsymbol{X}))], \tag{2}$$

where $\mathcal{L} : \text{supp}_Y \times \text{supp}_Y \rightarrow \mathbb{R}^+$ is a loss function. The following example illustrates these notions.

**Example 1** (Covariate shift). A common instance of the domain generalization problem considers source and target domains $\mathbb{M} : \{M^1, M^*\}$ over $\boldsymbol{V} = \{\boldsymbol{X}, Y\}$ and $\boldsymbol{U} = \{U_{\boldsymbol{X}}, U_Y\}$ defined by

$$\mathcal{M}^1 : \begin{cases} \mathcal{F}^1 : \begin{cases} \boldsymbol{X} \leftarrow f^1_{\boldsymbol{X}}(U_{\boldsymbol{X}}) \\ Y \leftarrow f_Y(\boldsymbol{X}, U_Y) \end{cases} \\ P^1(\boldsymbol{U}) = P^1(U_{\boldsymbol{X}}) \cdot P(U_Y) \end{cases} \qquad \mathcal{M}^* : \begin{cases} \mathcal{F}^* : \begin{cases} \boldsymbol{X} \leftarrow f^*_{\boldsymbol{X}}(U_{\boldsymbol{X}}) \\ Y \leftarrow f_Y(\boldsymbol{X}, U_Y) \end{cases} \\ P^*(\boldsymbol{U}) = P^*(U_{\boldsymbol{X}}) \cdot P(U_Y) \end{cases}$$

Here, because $P^1(U_{\boldsymbol{X}}) \neq P^1(U_{\boldsymbol{X}})$, this implies via Eq. 1 that the covariate distributions are different, i.e., $P^1(\boldsymbol{X}) \neq P^*(\boldsymbol{X})$. Still, the label distribution conditional on covariates is invariant, i.e., $P^1(Y \mid \boldsymbol{X}) = P^*(Y \mid \boldsymbol{X})$, also known as the covariate shift setting. Accordingly, the risk of a classifier $h := h(\boldsymbol{x})$ can be written as,

$$R_{P*}(h) = \int_{\text{supp}_Y \times \text{supp}_{\boldsymbol{X}}} \mathcal{L}(y, h(\boldsymbol{x})) P^*(y, \boldsymbol{x}) \cdot dy d\boldsymbol{x} = \int_{\text{supp}_Y \times \text{supp}_{\boldsymbol{X}}} \mathcal{L}(y, h(\boldsymbol{x})) P^1(y \mid \boldsymbol{x}) P^*(\boldsymbol{x}) \cdot dy d\boldsymbol{x}. \tag{3}$$

We will consider the problem of quantifying the variation in $R_{P*}(h)$ subject to variation in $P^*(\boldsymbol{x})$ that would be consistent with partial observations from these domains, e.g. samples from $P^1(\boldsymbol{X}, Y)$, and assumptions about the commonalities and discrepancies across the domains. □

To describe more general discrepancies in the mechanisms between the SCMs, we adapt the notion of domain discrepancy and selection diagram introduced in [19].

**Definition 2** (Domain discrepancy). For SCMs $\mathcal{M}^i, \mathcal{M}^j$ $(i, j \in \{*, 1, 2, \ldots, K\})$ defined over $\boldsymbol{V}$, the domain discrepancy set $\Delta_{ij} \subseteq \boldsymbol{V}$ is defined such that for every $V \in \Delta_{ij}$ there might exist a discrepancy $f_V^{\mathcal{M}^i} \neq f_V^{\mathcal{M}^j}$, or $P^{\mathcal{M}^i}(\boldsymbol{u}_V) \neq P^{\mathcal{M}^j}(\boldsymbol{u}_V)$. For abbreviation, we denote $\Delta_{i*}$ as $\Delta_i$. □

In words, if an endogenous variable $V$ is not in $\Delta_{ij}$, this means that the mechanism for $V$ (i.e., the function $f_V$ and the distribution of exogenous variables $P(\boldsymbol{u}_V)$) are structurally invariant across $\mathcal{M}^i, \mathcal{M}^j$. What follows integrates the domain discrepancy sets into a generalization of causal diagrams to express qualitative assumptions about multiple SCMs [26, 11].

**Definition 3** (Selection diagram). The selection diagram $\mathcal{G}^{\Delta_i}$ is constructed from $\mathcal{G}^i$ ($i \in \{1, 2, \ldots, T\}$) by adding the selection node $S_i$ to the vertex set, and adding the edge $S_i \to V$ for every $V \in \Delta_i$. The collection $\mathcal{G}^{\Delta} = \{\mathcal{G}^*\} \cup \{\mathcal{G}^{\Delta_i}\}_{i \in \{1, 2, \ldots, T\}}$ encodes the graphical assumptions. Whenever the causal diagram is shared across the domains, a single diagram can depict $\mathcal{G}^{\Delta}$. □

Selection diagrams extend causal diagrams and provide a parsimonious graphical representation of the commonalities and disparities across a collection of SCMs. The following example illustrates these notions and highlights various subtleties in the generalization error of different predictors.

**Example 2** (Generalization performance of classifiers). Consider the SCMs $\mathcal{M}^i$ ($i \in \{1, 2, *\}$) over the binary variables $\boldsymbol{X} = \{C_1, C_2, \ldots, C_{10}\} \cup \{W, Z\}$ and $Y$, defined as follows:

$$P^i(\boldsymbol{U}) : \begin{cases} \forall 1 \leqslant j \leqslant 10 : U_{C_j} \sim \text{Bern}(0.1) \text{ if } i = 1 \text{ Bern}(0.5) \text{ if } i = 2 \text{ Bern}(0.7) \text{ if } i = * \\ U_{YW} \sim \text{Bern}(0.2) \\ U_W \sim \text{Bern}(0.01) \text{ if } i = 1 \text{ Bern}(0.02) \text{ if } i = 2 \text{ Bern}(0.5) \text{ if } i = * \\ U_Z \sim \text{Bern}(0.9) \end{cases}$$

$$\mathcal{F}^i : \boldsymbol{C} \leftarrow \boldsymbol{U_C}, \; Y \leftarrow U_{YW} \oplus \bigoplus_{C \in \boldsymbol{C}} C, \; W \leftarrow U_{YW} \oplus U_W, \; Z \leftarrow Y \cdot U_Z + W \cdot (1 - U_Z)$$

$\oplus$ denotes the xor operator, *i.e.*, $A \oplus B$ evaluates to 1 if $A \neq B$ and evaluates to 0 if $A = B$. Notice that the distribution of exogenous noise associated with $\boldsymbol{C}_{1:10}$ and $\{W\}$ differs across the domains. Consider three baseline classifiers $h_1(\boldsymbol{c}, w) := w \oplus \bigoplus_{c \in \boldsymbol{c}} c$, $h_2(\boldsymbol{c}) := \bigoplus_{c \in \boldsymbol{c}} c$, $h_3(z) := z$ evaluated on data from $P^1, P^2, P^*$ with the symmetric loss function $\mathcal{L}(Y, h(\boldsymbol{X})) = \mathbb{1}\{Y \neq h(\boldsymbol{X})\}$. Their errors are given in Table 1. Notice that $h_1$ has almost perfect accuracy on both source distributions, but does not generalize to $\mathcal{M}^1$ as it uses the unstable feature $W$, incurring 50% loss. This observation indicates that mere minimization of the empirical risk might yield arbitrarily large risk in the unseen target domain. $h_2$ uses the features $\boldsymbol{C}$ that are the direct causes of $Y$, also known as the causal predictor [27, 2], and yields a stable loss of 20% across all domains. On the other hand, $h_3$ uses only $Z$ that is a descendant of $Y$, yet achieves a small loss across all domains as the mechanism of $Z$ is assumed to be invariant. This observation is surprising, because $h_3$ is neither a causal predictor nor the minimizer of the empirical risk, yet it performs nearly optimally on all domains. □

Example 2 illustrates potential challenges of the domain generalization problem, particularly regarding the variation of the risk of classifiers across the source and target domains. The following definition introduces the problem of "partial transportability" which is the main conceptual contribution of our paper. The objective is bounding a statistic of the target distribution using the data and assumptions available about related domains.

| Classifier | $R_{P\mathcal{M}^1}$ | $R_{P\mathcal{M}^2}$ | $R_{P\mathcal{M}*}$ |
|---|---|---|---|
| $h_1(\boldsymbol{c}, w)$ | 1% | 4% | 49% |
| $h_2(\boldsymbol{c})$ | 20% | 20% | 20% |
| $h_3(z)$ | 3% | 5% | 4% |

Table 1: Classifiers in Example 2.

**Definition 4** (Partial Transportability). Consider a system of SCMs $\mathbb{M} : \{\mathcal{M}^1, \mathcal{M}^2, \ldots, \mathcal{M}^K, \mathcal{M}^*\}$ that induces the selection diagram $\mathcal{G}^{\Delta}$ over the variables $\boldsymbol{V}$ and entails the distributions $\mathbb{P} : \{P^1(\boldsymbol{v}), P^2(\boldsymbol{v}), \ldots, P^K(\boldsymbol{v})\}$ and $P^*(\boldsymbol{v})$. A functional $\psi : \Omega_{\boldsymbol{V}} \to \mathbb{R}$ is partially transportable from $\mathbb{P}$ given $\mathcal{G}^{\Delta}$ if,

$$\mathbb{E}_{P\mathcal{M}_0^*}[\psi(\boldsymbol{V})] \leqslant q_{\max}, \forall \text{ SCMs } \mathbb{M}_0 \text{ that entail } \mathbb{P} \text{ and induce } \mathcal{G}^{\Delta}, \quad (4)$$

where $q_{\max} \in \mathbb{R}$ is a constant that can be obtained from $\mathbb{P}$ given $\mathcal{G}^{\Delta}$. □

For instance, finding the worst-case performance of a classifier based on the source distributions given the selection diagram is a special case of partial transportability with $\psi(\boldsymbol{x}, y) := \mathcal{L}(y, h(\boldsymbol{x}))$. In principle, this task is challenging as the exogenous distribution $P^*(\boldsymbol{U}_{\boldsymbol{V}})$ and structural assignments $f_V^*$ of variables $V \in \boldsymbol{V}$ that do not match with any of the source domains could be arbitrary. In the following section, we will define tractable parameterization of $\{P(\boldsymbol{U}), \mathcal{F}\}$ to derive a systematic approach to solving partial transportability tasks.

## 3 Canonical Models for Partial Transportability

We begin with an example to illustrate how one might approach parameterizing a query such as $\mathbb{E}_{P\mathcal{M}*}[\psi(\boldsymbol{V})]$, *e.g.*, the generalization error, to consistently solve the partial transportability task.

**Example 3** (The bow model). Let $\boldsymbol{X} := \{X\}$ be a single binary variable, and $Y$ be a binary label. Consider two source domains defined by the following SCMs:

$$\mathcal{M}^1 : \begin{cases} P^1(\boldsymbol{U}) : \begin{cases} U_X \sim \mathrm{Bern}(0.2) \\ U_Y \sim \mathrm{Bern}(0.05) \\ U_{XY} \sim \mathrm{Bern}(0.95) \end{cases} \\ \mathcal{F}^1 : \begin{cases} X \leftarrow U_X \oplus U_{XY} \\ Y \leftarrow (X \oplus U_{XY}) \oplus U_Y \end{cases} \end{cases} \qquad \mathcal{M}^2 : \begin{cases} P^2(\boldsymbol{U}) : \begin{cases} U_X \sim \mathrm{Bern}(0.9) \\ U_Y \sim \mathrm{Bern}(0.05) \\ U_{XY} \sim \mathrm{Bern}(0.95) \end{cases} \\ \mathcal{F}^2 : \begin{cases} X \leftarrow U_X \oplus U_{XY} \\ Y \leftarrow (X \oplus U_{XY}) \vee U_Y \end{cases} \end{cases}$$

The task is to evaluate the generalization error of the classifier $h(x) = \neg x$. $h$ can be shown optimal in both source domains: achieving $R_{P^1}(h) \approx 0.11$ and $R_{P^2}(h) \approx 0.06$. However, it is unclear whether it generalizes well to a target domain $\mathcal{M}^*$, given the domain discrepancy sets $\Delta_1 = \{X\}, \Delta_2 = \{Y\}$. □

Balke and Pearl [3] derived a canonical parameterization of SCMs such as $\{\mathcal{M}^1, \mathcal{M}^2, \mathcal{M}^*\}$ in Example 3. They showed that it is sufficient to parameterize $P(\boldsymbol{U})$ with correlated discrete latent variables $R_X, R_Y$, where $R_X$ determines the value of $X$, and $R_Y$ determines the functional that decides $Y$ based on $X$. The causal diagrams are shown in Figure

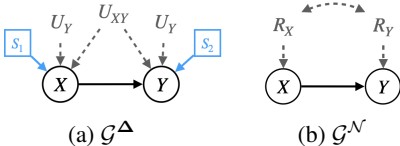

(a) $\mathcal{G}^\Delta$        (b) $\mathcal{G}^\mathcal{N}$

Figure 2: Selection diagram & Canonical param.

2. Canonical SCMs entails the same set of distributions as the true underlying SCMs, *i.e.* are equally expressive. In particular, Zhang and Bareinboim [42] showed that for every SCM $\mathcal{M}$, there exists an SCM of the described form specified with only a distribution $P(r_X, r_Y)$, where, $\mathrm{supp}_{R_X} = \{0, 1\}$, $\mathrm{supp}_{R_Y} = \{y = 0, y = 1, y = x, y = \neg x\}$. The joint distribution $P(r_X, r_Y)$ can be parameterized by a vector in 8-dimensional simplex, and entails all observational, interventional and counterfactual variables generated by the original SCM.

The following definition by Zhang et al. [42] provides a general formulation of canonical models.

**Definition 5** (Canonical SCM). A canonical SCM is an SCM $\mathcal{N} = \langle \boldsymbol{U}, \boldsymbol{V}, \mathcal{F}, P(\boldsymbol{U}) \rangle$ defined as follows. The set of endogenous variables $\boldsymbol{V}$ is discrete. The set of exogenous variables $\boldsymbol{U} = \{R_V : V \in \boldsymbol{V}\}$, where $\mathrm{supp}_{R_V} = \{1, \ldots, m_V\}$ and $m_V = |\{h_V : \mathrm{supp}_{pa_V} \to \mathrm{supp}_V\}|$. For each $V \in \boldsymbol{V}$, $f_V \in \mathcal{F}$ is defined as $f_V(pa_V, r_V) = h_V^{(r_V)}(pa_V)$.

**Example 3** (continued). Consider extending the canonical parameterization to to solve the partial transportability task by optimization. Each SCM $\mathcal{M}^1, \mathcal{M}^2, \mathcal{M}^*$ is associated with a canonical SCM $\mathcal{N}^1, \mathcal{N}^2, \mathcal{N}^*$. with exogenous variables $\{R_X, R_Y\}$ as above. The domain discrepancy sets $\boldsymbol{\Delta}$ indicate that certain causal mechanisms need to match across pairs of the SCMs. For example, $\Delta_1 = \{X\}$, which does not contain $Y$, and this means that (1) the function $f_Y$ is the same across $\mathcal{M}^1, \mathcal{M}^*$, and (2) the distribution of unobserved variables that are arguments of $f_Y$, namely, $U_{XY}, U_Y$ remains the same across $\mathcal{M}^1, \mathcal{M}^*$. Imposing these equalities on the canonical parameterization is straightforward as (1) the function $f_Y$ is the same across all canonical SCMs by construction, and (2) the only unobserved variable pointing to variable $V$ is $R_V$ (for $V \in \{X, Y\}$). Following the selection diagram shown in Fig. 2a, $\mathcal{M}^1, \mathcal{M}^*$ agree on the mechanism of $Y$, which translates to the constraint $P^{\mathcal{N}^1}(r_Y) = P^{\mathcal{N}^*}(r_Y)$. Similarly, $\mathcal{M}^2, \mathcal{M}^*$ agree on the mechanism of $X$ that translates to the constraint $P^{\mathcal{N}^2}(r_X) = P^{\mathcal{N}^*}(r_X)$. Putting these together, the optimization problem below finds the upper-bound for the risk $R_{P*}(h)$ for the classifier $h(x) = \neg x$:

$$\max_{\mathcal{N}^1, \mathcal{N}^2, \mathcal{N}*} P^{\mathcal{N}^*}(Y \neq \neg X) \tag{5}$$

$$\text{s.t. } P^{\mathcal{N}^1}(r_Y) = P^{\mathcal{N}^*}(r_Y), \quad P^{\mathcal{N}^2}(r_X) = P^{\mathcal{N}^*}(r_X) \qquad (Y \notin \Delta_1, \text{ and } X \notin \Delta_2)$$

$$P^{\mathcal{N}^1}(x, y) = P^1(x, y), \quad P^{\mathcal{N}^2}(x, y) = P^2(x, y) \qquad \text{(matching source dists)}$$

Notably, the above optimization has a linear objective with linear equality constraints. □

This example illustrates a more general strategy, in which probabilities induced by an SCM over discrete endogenous variables $\boldsymbol{V}$ may be generated by a canonical model. What follows is the main result of this section, and provides a systematic approach to partial transportability using the canonical models.

**Theorem 1** (Partial-TR with canonical models). *Consider the tuple of SCMs $\mathbb{M}$ that induces the selection diagram $\mathcal{G}^{\boldsymbol{\Delta}}$ over the variables $\boldsymbol{V}$, and entails the source distributions $\mathbb{P}$, and the target distribution $P^*$. Let $\psi : \Omega_{\boldsymbol{V}} \to \mathbb{R}$ be a functional of interest. Consider the following optimization scheme:*

$$\max_{\mathcal{N}^1, \mathcal{N}^2, \ldots, \mathcal{N}*} \mathbb{E}_{P^{\mathcal{N}*}} [\psi(\boldsymbol{V})] \; s.t. \; P^{\mathcal{N}^i}(\boldsymbol{v}) = P^i(\boldsymbol{v}) \qquad \forall i \in \{1, 2, \ldots, K, *\} \tag{6}$$

$$P^{\mathcal{N}^i}(r_V) = P^{\mathcal{N}^j}(r_V), \quad \forall i, j \in \{1, 2, \ldots, K, *\} \quad \forall V \notin \Delta_{i,j}$$

*where each $\mathcal{N}^i$ is a canonical model characterized by a joint distribution over $\{R_V\}_{V \in \boldsymbol{V}}$. The value of the above optimization is a tight upper-bound for the quantity $\mathbb{E}_{P*}[\psi(\boldsymbol{V})]$ among all tuples of SCMs that induce $\mathcal{G}^{\boldsymbol{\Delta}}$ and entail $\mathbb{P}$.* □

In words, this Theorem states that one may tightly bound the value of a target quantity $\mathbb{E}_{P*}[\psi(\boldsymbol{V})]$ by optimizing over the space of canonical models subject to the proposed constraints, without any loss of information. An implementation of Thm. 1 approximating the worst-case error, by making inference on the posterior distribution of the target quantity, is provided in Appendix A.

## 4 Neural Causal Models for Partial Transportability

In this section we consider inferences in more general settings by using neural networks as a generative model, acting as a proxy for the underlying SCMs $\mathbb{M}$ with the potential to scale to real-world, high-dimensional settings while preserving the validity and tightness of bounds. For this purpose, we consider Neural Causal Models [41] and adapt them for the partial transportability task. What follows is an instantiation of [41, Definition 7].

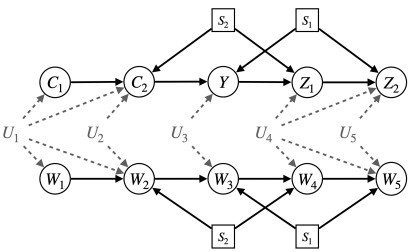

Figure 3: Selection diagram for Example 4.

**Definition 6** (Neural Causal Model). A Neural Causal Model (NCM) corresponding to the causal diagram $\mathcal{G}$ over the discrete variables $\boldsymbol{V}$ is is an SCM defined by the exogenous variables:

$$\boldsymbol{U} = \{U_{\boldsymbol{W}} \sim \mathrm{unif}(0, 1) : \boldsymbol{W} \subseteq \boldsymbol{V} \; \text{s.t.} \; A \leftrightarrow B \in \mathcal{G}, \quad \forall A, B \in \boldsymbol{W}\}, \tag{7}$$

and the functional assignments $V \leftarrow f_{\theta_V}(Pa_V, \boldsymbol{U}_V)$, where $\boldsymbol{U}_V = \{U_{\boldsymbol{W}} \in \boldsymbol{U} : V \in \boldsymbol{W}\}$. The function $f_{\theta_V}$ is a feed-forward neural network parameterized with $\theta_V$ that outputs in $\mathrm{supp}_V$. The distribution entailed by an NCM is denoted by $P(\boldsymbol{v}; \theta)$, where $\theta = \{\theta_V\}_{V \in \boldsymbol{V}}$.

To illustrate how one might leverage this parameterization to define an instance of partial transportability task consider the following example.

**Example 4.** Let SCMs $\mathcal{M}^1, \mathcal{M}^2, \mathcal{M}^*$ induce $\mathcal{G}^{\boldsymbol{\Delta}}$ shown in Fig. 3 over the binary variables $\boldsymbol{X}, Y$, where $\boldsymbol{X} = \{C_1, C_2, Z_1, Z_2, W_1, \ldots, W_5\}$. Let $\theta^1, \theta^2, \theta^*$ be the parameters of NCMs constructed based on the causal diagram in Fig. 3 (without the s-nodes). The objective is to constrain these parameters to simulate a compatible tuple of NCMs $\mathcal{M}_{\theta^1}, \mathcal{M}_{\theta^2}, \mathcal{M}_{\theta*}$ that equivalently entail $P^1(\boldsymbol{x}, y), P^2(\boldsymbol{x}, y)$ and induce $\mathcal{G}^{\boldsymbol{\Delta}}$.

For instance, the fact that $S_2$ is not pointing to $Y$ suggests the invariance $f_Y^* = f_Y^2$ and $P^*(\boldsymbol{u}_Y) = P^2(\boldsymbol{u}_Y)$ for the true underlying SCMs. That same invariance may be enforced in the corresponding NCMs by relating the parameterization of $\mathcal{M}_{\theta^2}, \mathcal{M}_{\theta*}$, i.e., imposing that $\theta_Y^* = \theta_Y^2$ for the NN generating $Y$. Similarly, the observed data $D^1, D^2$ from the source distributions $P^1(\boldsymbol{x}, y), P^2(\boldsymbol{x}, y)$, respectively, impose constraints on the parameterization of NCMs as plausible models must satisfy $P(\boldsymbol{x}, y; \theta^1) = P^1(\boldsymbol{x}, y)$ and $P(\boldsymbol{x}, y; \theta^2) = P^2(\boldsymbol{x}, y)$. This may be enforced, for instance, by maximizing the likelihood of data w.r.t. the NCM parameters: $\theta^i \in \arg\max_\theta \sum_{\boldsymbol{x}, y \in D^i} \log P(\boldsymbol{x}, y; \theta^i)$, for $i \in \{1, 2\}$. By extending this intuition for all constraints imposed by the selection diagram and data, we narrow the set of NCMs $\mathcal{M}_{\theta^1}, \mathcal{M}_{\theta^2}, \mathcal{M}_{\theta*}$ to a set that is compatible with our assumptions and data. Maximizing the risk of some prediction function $R_{P*}(h)$ in this class of constrained NCMs might then achieve an informative upper-bound. □

Motivated by the observation in Example 4, we now show a more formal result (analogous to Thm. 1) that guarantees that the solution to the partial transportability task in the space of constrained NCMs achieves a tight bound on a given target quantity $\mathbb{E}_{P*}[\psi(\boldsymbol{V})]$.

**Theorem 2** (Partial-TR with NCMs). *Consider a tuple of SCMs $\mathbb{M}$ that induces $\mathcal{G}^{\Delta}, \mathbb{P}$ and $P*$ over the variables $\boldsymbol{V}$. Let $D^i \sim P^i(x, y)$ denote the samples drawn from the $i$-th source domain. Let $\theta^i$ denote the parameters of NCM corresponding to $\mathcal{M}^i \in \mathbb{M}$. Let $\mathbb{E}_{P*}[\psi(\boldsymbol{V})]$ be the target quantity. The solution to the optimization problem,*

$$\hat{\Theta} \in \underset{\Theta:\langle \theta^1, \theta^2, \ldots, \theta^K, \theta* \rangle}{\arg\max} \sum_{\boldsymbol{w}} \psi(\boldsymbol{w}) \cdot \sum_{\boldsymbol{v} \backslash \boldsymbol{w}} P(\boldsymbol{v}; \theta*) \tag{8}$$

$$s.t. \; \theta_V^i = \theta_V^j, \qquad \forall i, j \in \{1, 2, \ldots, K, *\} \quad \forall V \notin \Delta_{i,j}$$

$$\theta^i \in \underset{\theta}{\arg\max} \sum_{\boldsymbol{v} \in D^i} \log P(\boldsymbol{v}; \theta), \quad \forall i \in \{1, 2, \ldots, K\}.$$

*is a tuple of NCMs that induce $\mathcal{G}^{\Delta}$, entails $\mathbb{P}$. In the large sample limit, the solution yields a tight upper-bound for $\mathbb{E}_{P*}[\psi(\boldsymbol{V})]$.* □

Theorem 2 establishes the expressive power of NCMs for solving partial transportability tasks. This formulation is powerful because it enables the use of gradient-based optimization of neural networks for learning and, in principle, might scale to large number of variables.

## 4.1 Neural-TR: An Efficient Implementation

We could further explore the efficient optimization of parameters by exploiting the separation between variables in the selection diagram. Rahman et al. [28], for instance, show that the NCM parameterization is modular w.r.t. the c-components of the causal diagram. We can similarly elaborate on this property, and leverage it for more efficient partial transportability.

In the following, we build towards an efficient algorithm for partial transportability using NCMs by first showing an example that describes how a given target quantity $\mathbb{E}_{P*}[\psi(\boldsymbol{V})]$ might be decomposed for learning more efficiently.

**Example 4** (continued). $P(\boldsymbol{x}, y; \theta*)$ in the objective in Eq. (8) may be decomposed as follows:

$$P(\boldsymbol{x}, y; \theta*) = P*(\underbrace{c_1, c_2, w_1, w_2}_{\boldsymbol{a}_1}; \theta_{\boldsymbol{A}_1}*) \cdot P(\underbrace{y, w_3}_{\boldsymbol{a}_2} \mid \underbrace{c_2, w_2}_{\boldsymbol{b}_2}; \theta_{\boldsymbol{A}_2}*) \cdot P(\underbrace{z_1, z_2, w_4, w_5}_{\boldsymbol{a}_3} \mid \underbrace{y, w_3}_{\boldsymbol{b}_3}; \theta_{\boldsymbol{A}_3}*),$$

where the subsets $\boldsymbol{A}_1, \boldsymbol{A}_2, \boldsymbol{A}_3$ are the $c$-components of $\mathcal{G}*$. Notice, $S_2$ is not pointing to any of the variables $\boldsymbol{A}_2$, which means that their mechanism is shared across $\mathcal{M}^2, \mathcal{M}*$, and therefore,

$$P(\boldsymbol{a}_2 \mid \boldsymbol{b}_2; \theta_{\boldsymbol{A}_2}*) = P(\boldsymbol{a}_2 \mid \boldsymbol{b}_2; \theta_{\boldsymbol{A}_2}^2) \approx P^2(\boldsymbol{a}_2 \mid \boldsymbol{b}_2). \tag{9}$$

This property is the basis of transportability algorithms [4, 10], and is known as the s-admissibility criterion [26], which allows us to deduce distributional invariances from structural invariances. By Eq. (9), we can replace the term $P(\boldsymbol{a}_2 \mid \boldsymbol{b}_2; \theta_{\boldsymbol{A}_2}*)$ in the objective with the probabilistic model $P(\boldsymbol{a}_2 \mid \boldsymbol{b}_2; \eta^2)$ that is trained with $D^2$ to approximate $P^2(\boldsymbol{a}_2 \mid \boldsymbol{b}_2)$ and plug it into the objective $Eq.$ (8) as a constant.

As a consequence, we do not need to optimize over the parameters $\theta_{\boldsymbol{A}_2}^1, \theta_{\boldsymbol{A}_2}^2, \theta_{\boldsymbol{A}_2}*$ from the partial transportability optimization problem. Similarly, since $S_1$ does not point to $\boldsymbol{A}_1$, we can substitute $P(\boldsymbol{a}_1; \theta*)$ with $P(\boldsymbol{a}_1; \eta^1)$, and pre-train it with data $D^1$. In the context of Example 4 and the evaluation of $R_{P*}(h)$, the objective in Eq. (8) may be simplified to the substantially lighter optimization task:

$$\max_{\theta_{\boldsymbol{A}_3}^1, \theta_{\boldsymbol{A}_3}^2, \theta* \boldsymbol{A}_3} \mathbb{E}_{\boldsymbol{A}_1 \sim P(\boldsymbol{a}_1; \eta^1)} \Big[ \mathbb{E}_{\boldsymbol{A}_2 \sim P(\boldsymbol{a}_2 \mid \boldsymbol{b}_2; \eta^2)} \big[ \sum_{\boldsymbol{a}_3} P(\boldsymbol{a}_3 \mid \boldsymbol{b}_3; \theta_{\boldsymbol{A}_3}*) \cdot \mathbb{1}\{h(\boldsymbol{a}_1, \boldsymbol{a}_2, \boldsymbol{a}_3 \backslash \{y\}) \neq y\} \big] \Big]$$

$$s.t. \; \theta_{\boldsymbol{A}_3}^i \in \underset{\theta_{\boldsymbol{A}_3}}{\arg\max} \sum_{\boldsymbol{a}_3, \boldsymbol{b}_3 \in D^i} \log P(\boldsymbol{a}_3 \mid \boldsymbol{b}_3; \theta_{\boldsymbol{A}_3}), \quad \text{for } i \in \{1, 2\}. \tag{10}$$

In general, the parameter space of NCMs can be cleverly decoupled and the computational cost of the optimization problem can be significantly improved since only a subset of the conditional distributions need to be parameterized and optimized. This observation motivates Alg. 1 designed to exploit these insights. It proceeds by first, decomposing the query, second, computing the identifiable components, and third, parameterizing the components that are not point identifiable and running the NCM optimization routine. The following proposition demonstrates the correctness of this procedure.

---

**Algorithm 1** Neural-TR

---

**Require:** Source data $D^1, D^2, \ldots, D^K$; selection diagram $\mathcal{G}^{\boldsymbol{\Delta}}$; functional $\psi : \Omega_{\boldsymbol{W}} \to [0,1]$.
**Ensure:** Upper-bound for $\mathbb{E}_{P*}[\psi(\boldsymbol{W})]$
1: $\{\boldsymbol{A}_j\}_{j=1}^m \leftarrow$ c-components of $\boldsymbol{A} := \mathbf{An}_{\mathcal{G}*}(\boldsymbol{W})$ in causal diagram $\mathcal{G}^*$.
2: $\Theta, \mathbb{C}_{\text{expert}} \leftarrow \varnothing, \quad \mathcal{L}_{\text{data}} \leftarrow 0$
3: $P^*(\boldsymbol{w}) := \sum_{\boldsymbol{a} \backslash \boldsymbol{w}} \prod_{j=1}^m P^*(\boldsymbol{a}_j \mid \mathrm{do}(pa_{\boldsymbol{A}_j}))$
4: **for** $j = 1$ to $m$ **do**
5:     **if** $\exists i \in \{1, 2, \ldots, K\}$ such that $\boldsymbol{A}_j \cap \Delta_{*i} = \varnothing$ **then**
6:         $\eta_{\boldsymbol{A}_j}^i \leftarrow \arg\max_{\eta_{\boldsymbol{A}_j}} \sum_{\boldsymbol{a}_j, pa_{\boldsymbol{A}_j} \in D^i} \log P(\boldsymbol{a}_j \mid \mathrm{do}(pa_{\boldsymbol{A}_j}); \eta_{\boldsymbol{A}_j})$
7:         In $P^*(\boldsymbol{w})$, replace $P^*(\boldsymbol{a}_j \mid \mathrm{do}(pa_{\boldsymbol{A}_j}))$ with $P(\boldsymbol{a}_j \mid \mathrm{do}(pa_{\boldsymbol{A}_j}); \eta_{\boldsymbol{A}_j}^i)$.
8:     **else**
9:         $\Theta \leftarrow \Theta \cup \{\theta_{\boldsymbol{A}_j}^i\}_{i \in \{1,2,\ldots,K,*\}}$
10:       In $P^*(\boldsymbol{w})$, replace $P^*(\boldsymbol{a}_j \mid \mathrm{do}(pa_{\boldsymbol{A}_j}))$ with $P(\boldsymbol{a}_j \mathrm{do}(pa_{\boldsymbol{A}_j}); \theta_{\boldsymbol{A}_j}^*)$.
11:       $\mathbb{C}_{\text{expert}} \leftarrow \mathbb{C}_{\text{expert}} \cup \{\{\theta_V^i = \theta_V^*\}_{V \in \boldsymbol{A}_j \backslash \Delta_{*i}}\}_{i=1}^K$
12:       $\mathcal{L}_{\text{likelihood}} \leftarrow \mathcal{L}_{\text{likelihood}} + \sum_{\boldsymbol{a}_j, pa_{\boldsymbol{A}_j} \in D^i} \log P(\boldsymbol{a}_j, \mathrm{do}(pa_{\boldsymbol{A}_j}); \theta_{\boldsymbol{A}_j}^i)$.
13:     **end if**
14: **end for**
15: **Return** $\hat{\Theta} \leftarrow \arg\max_{\Theta} \sum_{\boldsymbol{w}} P^*(\boldsymbol{w}; \Theta) \cdot \psi(\boldsymbol{w}) + \Lambda \cdot \mathcal{L}_{\text{likelihood}}(\Theta)$ subject to $\mathbb{C}_{\text{expert}}$

---

**Proposition 1.** *Neural-TR (Algorithm 1) computes a tuple of NCMs compatible with the source data and graphical assumptions that yields the upper-bound for $\mathbb{E}_{P*}[\psi(\boldsymbol{W})]$ in the large sample limit.* □

This result may be understood as an enhancement of Thm. 2 in which the factors that are readily transportable from source data are taken care of in a pre-processing step. The hybrid approach is especially useful in case researchers have pre-trained probabilistic models with arbitrary architecture that they can use off-the-shelf and avoid unnecessary computation.

### 4.2 Neural-TR for the Optimization of Classifiers

The Neural-TR algorithm can be viewed as an adversarial domain generator that takes a classifier $h(\boldsymbol{z})$ as the input, and then parameterizes a collection of SCMs to find a plausible target domain that yields the worst-case risk for the given classifier, namely, $\hat{\theta}^*$. By flipping $h(\boldsymbol{z})$ for some $\boldsymbol{z} \in \Omega$ we can reduce the risk of $h$ under $\hat{\theta}^*$.

Interestingly, we can exploit Neural-TR to generate adversarial data for a given classifier and introduce an iterative procedure to progressively train classifiers with with minimum risk upper-bound. Algorithm 2 describes this approach. At each iteration, CRO uses Neural-TR as a subroutine to obtain an adversarially designed NCM $\hat{\theta}^*$ that yields the worst-case risk for the classifier at hand. Next, it collects data $D^*$ from this NCM and adds it to a collection of datasets $\mathbb{D}^*$. Finally, it updates the classifier to be robust to the collection $\mathbb{D}^*$ by minimizing

---

**Algorithm 2** CRO (Causal Robust Optimization)

---

**Require:** $\mathbb{D} : \langle D^1, D^2, \ldots, D^K \rangle; \mathcal{G}^{\boldsymbol{\Delta}}; \delta > 0$
**Ensure:** $h(\boldsymbol{X})$ with the best worst-case risk.
1: Initialize $h$ randomly and $\mathbb{D}^* \leftarrow \varnothing$
2: $\hat{\Theta} \leftarrow$ Neural-TR$(\mathbb{D}, \mathcal{G}^{\boldsymbol{\Delta}}, \psi : \mathcal{L}(h(\boldsymbol{x}), y))$
3: **while** $R_{P(\boldsymbol{x},y;\hat{\theta}*)}(h) - \max_{D \in \mathbb{D}*} R_D(h) > \delta$ **do**
4:     $\mathbb{D}^* \leftarrow \mathbb{D}^* \cup \{D^* \sim P(\boldsymbol{x}, y; \hat{\theta}^*)\}$
5:     $h \leftarrow \arg\min_h \max_{D \in \mathbb{D}*} R_D(h)$
6:     $\hat{\Theta} \leftarrow$ Neural-TR$(\mathbb{D}, \mathcal{G}^{\boldsymbol{\Delta}}, \psi : \mathcal{L}(h(\boldsymbol{x}), y))$
7: **end while**
8: **Return** $h$

---

the maximum of the empirical risk $R_D(h) := \sum_{\boldsymbol{x}, y \in D} \mathcal{L}(y, h(x))$ across all $D \in \mathbb{D}^*$. We repeat this process until convergence of the upper-bound for risk. The following result justifies optimality of CRO for domain generalization; more discussion is provided in Appendix C.2.

**Theorem 3** (Domain generalization with CRO)**.** *Algorithm 2 returns a worst-case optimal solution;*

$$\mathrm{CRO}(\mathbb{D}, \mathcal{G}^{\boldsymbol{\Delta}}) \in \underset{h:\Omega_{\boldsymbol{X}} \to \Omega_Y}{\arg\min} \quad \underset{\text{tuple of SCMs } \mathbb{M}_0 \text{ that entails } \mathbb{P} \text{ \& induces } \mathcal{G}^{\boldsymbol{\Delta}}}{\max} R_{P^{\mathcal{M}_0^*}}(h). \tag{11}$$

In words, Thm. 3 states that the classifier returned by CRO, in the large sample limit, minimizes worst-case risk in the target domain subject to the constraints entailed by the available data and induced by the structural assumptions.

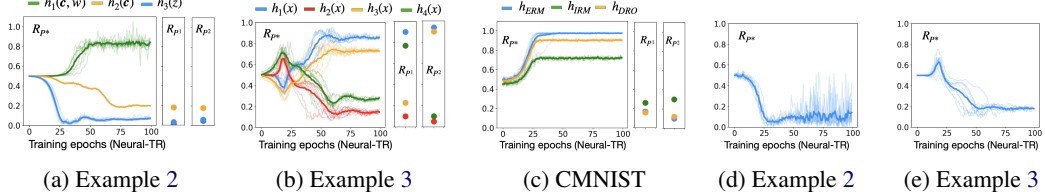

| (a) Example 2 | (b) Example 3 | (c) CMNIST | (d) Example 2 | (e) Example 3 |

Figure 4: (a-c): worst-case risk evaluation results as a function of Neural-TR (Alg. 1) training iterations. (d,e): worst-case risk evaluation of CRO.

## 5 Experiments

This section illustrates Algs. 1 and 2 for the evaluation and optimization of the generalization error on several tasks, ranging from simulated examples to semi-synthetic image datasets. The details of the experimental set-up and examples not fully described below, along with additional experiments, can be found in the Appendix.

### 5.1 Simulations

**Worst-case risk evaluation**  Our first experiment revisits Examples 2 and 3 for the evaluation of the worst-case risk $R_{P*}$ of various classifiers with Neural-TR (Alg. 1).

In Example 2 we had made (anecdotal) performance observations for the classifiers $h_1(\boldsymbol{c}, w) := w \oplus \bigoplus_{c \in \boldsymbol{c}} c, h_2(\boldsymbol{c}) := \bigoplus_{c \in \boldsymbol{c}} c, h_3(z) := z$ in a selected target domain $\mathcal{M}^*$. We now consider providing a worst-case risk *guarantee* with Neural-TR for *any* (compatible) target domain. The main panel in Fig. 4a shows the convergence of the worst-case risk evaluator over successive training iterations (line 15, Alg. 1), repeated 10 times with different model seeds and solid lines denoting the mean worst-case risk. The source performances $R_{P_1}, R_{P_2}$ are given in the two right-most panels for reference. We observe that the good source performance of $h_2(\boldsymbol{c})$ and $h_3(z)$ generalizes to *all* possible target domains consistent with our assumptions, while the classifier $h_1(\boldsymbol{c}, w)$ diverges, with an error of $90\%$ in the worst target domain. In Example 3, we consider the evaluation of binary classifiers $h \in \{h_1(x) := x, h_2(x) := \neg x, h_3(x) := 0, h_4(x) := 1\}$. $h_2(x) = \neg x$. Our results are given in Fig. 4b, highlighting the extent to which source performance need not be indicative of target performance. With these results, we are now in a position to confirm the desirable performance profile of $h_2$, even in the worst-case, as hypothesized in Example 3.

**Worst-case risk optimization**  For each one of the examples above, we implement CRO (Alg. 2) to uncover the theoretically optimal classifier in the worst-case. The worst-case risks of the classifiers learned by CRO, denoted $h_{\mathrm{CRO}}$, are given by $0.05$ for Example 2 and $0.18$ for Example 3. The worst-case risk evaluation results (with Neural-TR, as above) are given in Figs. 4d and 4e. It is interesting to note that these errors coincide with the best performing classifiers considered in the previous experiment, i.e. $h_3(z) := z$ for Example 2 and $h_2(x) = \neg x$ for Example 3. In fact, by comparing the outputs of CRO $h_{\mathrm{CRO}}$ with these classifiers, we can verify that the classifiers learned by CRO in these examples are precisely the mappings $h_{\mathrm{CRO}}(z) := z$ and $h_{\mathrm{CRO}}(x) = \neg x$ which is remarkable. By Thm. 3, $h_3(z) := z$ and $h_2(x) = \neg x$ are the theoretically best worst-case classifiers among all possible functions given the data and assumptions.

### 5.2 Colored MNIST

Our second experiment considers the colored MNIST (CMNIST) dataset that is used in the literature to highlight the robustness of classifiers to spurious correlations, e.g. see [2]. The goal of the classifier is to predict a binary label $Y \in \{0, 1\}$ assigned to an image $\boldsymbol{Z} \in \mathbb{R}^{28 \times 28 \times 3}$ based on whether the digit in the image is greater or equal to five. MNIST images $\boldsymbol{W} \in \mathbb{R}^{28 \times 28}$ are grayscale (and latent), and color $C \in \{\mathrm{red}, \mathrm{green}\}$ correlates with the class label $Y$.

Following standard implementations, we construct datasets from three domains with varying correlation strength between the color and image label: set to $90\%$ agreement between the color $C = \mathrm{red}$ and label $Y = 1$ in source domain $\mathcal{M}^1$, and $80\%$ in source domain $\mathcal{M}^2$. We consider performance evaluation and optimization in a target domain $\mathcal{M}^*$ with potential discrep-

Figure 5: $\mathcal{G}^{\triangle}_{\mathrm{CMNIST}}$

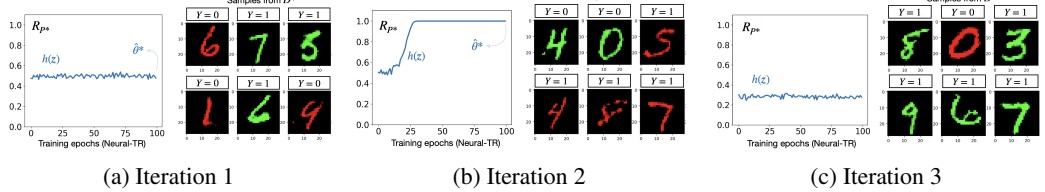

| (a) Iteration 1 | (b) Iteration 2 | (c) Iteration 3 |

Figure 6: Illustration of the CRO training process (Alg. 2) on the colored MNIST task.

ancies in the mechanism for $C$, rendering the correlation between color and label unstable. The selection diagram is given in Figure 5.

**Worst-case risk evaluation**    Consider a setting in which we are given a classifier $h : \Omega_{\boldsymbol{Z}} \to \Omega_Y$, and the task is to assess its generalizability with a symmetric 0-1 loss function. We use data drawn from $P^{1,2}(\boldsymbol{z}, y)$ to train predictors using Empirical Risk Minimization (ERM) [38], Invariant Risk Minimization (IRM) [2], and group Distributionally Robust Optimization (group DRO) [32], namely $h_{\mathrm{ERM}}(\boldsymbol{z}), h_{\mathrm{IRM}}(\boldsymbol{z})$, and $h_{\mathrm{DRO}}(\boldsymbol{z})$ respectively; more detailed discussion about the role of invariance and robustness in domain generalization is available in appendix D. Using Neural-TR, we observe in Fig. 4c that the worst-case risk of $h_{\mathrm{ERM}}$ in a target domain with a discrepancy in the color assignment is approximately $0.95$, $h_{\mathrm{DRO}}$ achieves $0.90$ worst-case risk, and $h_{\mathrm{IRM}}$ achieves $0.65$ worst-case risk. Either method perform worse than the baseline, that is random classification with risk $0.5$. On this task, a classifier trained on gray-scale images $\boldsymbol{W}$ achieves a worst-case error of $0.25$.

**Worst-case risk optimization**    We now ask whether we could learn a theoretically optimal classifier in the worst-case with CRO (Alg. 2). Fig. 6 illustrates the training process over several iterations. Specifically, given a randomly initialized $h$, we infer the NCM $\hat{\theta}^*$ that entails worst-case performance of $h$ (in this case, chance performance $R_{P*}(h) = 0.5$) and generate data $D^*$ from $\hat{\theta}^*$, shown in Fig. 6a. In a second iteration, a new candidate $h$ is trained to minimize worst-case risk on $\mathbb{D} = D^*$. Note that in $D^*$, we observe an almost perfect association between the color $C$ =green and label $Y = 1$: $h$ therefore is encouraged to exploit color for prediction. Its worst-case error (inferred with Neural-TR) is accordingly close to 1, and the corresponding worst-case NCM $\hat{\theta}^*$ entails a distribution of data in which the correlation between color and label is flipped: with a strong association between the color $C$ =red and label $Y = 1$, as shown in Fig. 6b. In a third iteration, a new candidate $h$ is trained to minimize worst-case risk on the updated $\mathbb{D}^*$ with data samples from the previous two iterations (exhibiting opposite color-label correlations). By construction, this classifier is trained to ignore the spurious association between color and label, classifying images based on the inferred digit which leads to better behavior in the worst-case: achieving a final error of approximately $0.25$, as shown in Fig. 6c, which is theoretically optimal. Note, however, that the poor performance of the baseline algorithms is not directly comparable to that of CRO, since CRO has access to background information (selection diagrams) that can not be communicated with the baseline algorithms. CRO may thus be interpreted as a meta-algorithm that operates with a broader range of assumptions encoded in a certain format (i.e., the selection diagram) that enable it to find the theoretically optimal classifier for domain generalization, in contrast to the baseline algorithms.

## 6    Conclusion

Guaranteeing the performance of ML algorithms implemented in the wild is a critical ingredient for improving the safety of AI. In practice, evaluating the performance of a given algorithm is non-trivial. Often the performance may vary as a consequence of our uncertainty about the possible target domain, also called a non-transportable setting. In this paper, we provide the first general estimation technique for bounding an arbitrary statistic such as the classification risk across multiple domains. More specifically, we extend the formulation of canonical models and neural causal models for the transportability task, demonstrating that tight bounds may be estimated with both approaches. Building on these theoretical findings, we introduce a Bayesian inference procedure as well as a gradient-based optimization algorithm for scalable inferences in practice. Moreover, we introduce Causal Robust Optimization (CRO), an iterative learning scheme that uses partial transportability as a subroutine to find a predictor with the best worst-case risk given the data and graphical assumptions.

## 7 Acknowledgement

This research was supported in part by the NSF, ONR, AFOSR, DARPA, DoE, Amazon, JP Morgan, and The Alfred P. Sloan Foundation.

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

# Appendix

## Table of Contents

# A  Partial Transportability as a Bayesian Inference Task

Consider a system of multiple SCMs $\mathcal{M}^1, \mathcal{M}^2, \ldots, \mathcal{M}^K, \mathcal{M}^*$ that induces the selection diagram $\mathcal{G}^{\Delta}$, and entails the source distributions $P^1, P^2, \ldots, P^K$, and the target distribution $P^*$ over the variables $\boldsymbol{V}$. Let $\psi : \Omega_{\boldsymbol{X}} \to \mathbb{R}$ be a functional of interest. Consider the following optimization scheme:

$$\hat{q}_{\max} = \max_{\mathcal{N}^1, \mathcal{N}^2, \ldots, \mathcal{N}^*} \mathbb{E}_{P^{\mathcal{N}*}}[\psi(\boldsymbol{X})] \tag{12}$$

$$\text{s.t. } P^{\mathcal{N}^i}(r_V) = P^{\mathcal{N}^j}(r_V), \qquad \forall i, j \in \{1, 2, \ldots, K, *\} \quad \forall V \notin \Delta_{i,j}$$

$$P^{\mathcal{N}^i}(\boldsymbol{v}) = P^i(\boldsymbol{v}) \qquad \forall i \in \{1, 2, \ldots, K, *\},$$

where each $\mathcal{N}^i$ is a canonical model characterized by a joint distribution over $\{R_V\}_{V \in \boldsymbol{V}}$.

This section describes an Markov Chain Monte Carlo (MCMC) algorithm to approximate the optimal scalar $\hat{q}_{\max}$ upper bounding the query $\phi_{\mathcal{N}*} := \mathbb{E}_{P^{\mathcal{N}*}}[\psi(\boldsymbol{X})]$ above from finite samples drawn from input distributions $P^1, P^2, \ldots, P^K$. Formally, we aim to infer the value,

$$\hat{q}_{\max} : P(\phi_{\mathcal{N}*} < \hat{q}_{\max} \mid \bar{\boldsymbol{v}}) = 1 \tag{13}$$

where $\bar{\boldsymbol{v}} := (\bar{\boldsymbol{v}}_{P^1}, \ldots, \bar{\boldsymbol{v}}_{P^k})$, $\bar{\boldsymbol{v}}_{P^i} = \{\boldsymbol{v}_{P^i}^{(j)} : j = 1, \ldots, n_i\}$ denote $n_i$ independent sampled drawn from $P^i$.

We consider a setting in which we are provided with prior distributions (possibly uninformative) over parameters of the family of compatible CMs $\mathcal{N}^1, \mathcal{N}^2, \ldots, \mathcal{N}^*$. In particular, we assume that for each CM, probabilities of $P(U), U \in \boldsymbol{U}$ are drawn from uninformative Dirichlet priors; and $\mathcal{F}$ are drawn uniformly from the finite class of possible structural functions. That is, for every $U \in \boldsymbol{U}$ and every $V \in \boldsymbol{V}$,

$$P(U) \sim \texttt{Dirichlet}(\alpha_1, \ldots, \alpha_{d_U}), \quad f_V \sim \texttt{Uniform}(\Omega_{PA_V} \times \Omega_{\boldsymbol{U}_V} \mapsto \Omega_V) \tag{14}$$

where $d_U = \prod_{V \in Pa(\boldsymbol{C}_U)} |\Omega_V|$ and $\alpha_1 = \ldots = \alpha_{d_U} = 1$.

The total collection of parameters is given by the set $\{(\boldsymbol{\theta}^{\mathcal{N}^1}, \boldsymbol{\xi}^{\mathcal{N}^1}), \ldots, (\boldsymbol{\theta}^{\mathcal{N}^*}, \boldsymbol{\xi}^{\mathcal{N}^*})\}$. Among them $\boldsymbol{\theta} = \{\boldsymbol{\theta}_U \in [0,1]^{d_U} : U \in \boldsymbol{U}\}$ define the parameterization of exogenous probabilities while $\boldsymbol{\xi} = \{\xi_V^{(pa_V, \boldsymbol{u}_V)} \in \text{supp}_V : PA_V \subset \boldsymbol{V}, \boldsymbol{U}_V \subset \boldsymbol{U}\}$ define the structural functions, one set of each CM separately.

We design a Gibbs sampler to evaluate posterior distributions over these parameters. For simplicity, we describe each step of the gibbs sampler for a single domain and input dataset, and consider the implementation of constraints below.

## A.1  Gibbs Sampling

The Gibbs sampler iterates over the following steps, each parameter conditioned on the current values of the remaining terms in the parameter vector.

1. *Sample $\boldsymbol{u}$.* Let $u \in \Omega_U, U \in \boldsymbol{U}$. For each observed data example across all domains $\boldsymbol{v}^{(n)} \in \bar{\boldsymbol{v}}$, $n = 1, \ldots, \sum_i n_i$, we sample corresponding exogeneous variables $U \in \boldsymbol{U}$ from the conditional distribution,

$$P(\boldsymbol{u}^{(n)} \mid \boldsymbol{v}^{(n)}, \boldsymbol{\xi}, \boldsymbol{\theta}) \propto P(\boldsymbol{u}^{(n)}, \boldsymbol{v}^{(n)} \mid \boldsymbol{\xi}, \boldsymbol{\theta}) = \prod_{V \in \boldsymbol{V}} \mathbb{1}\{\xi_V^{(pa_V^{(n)}, \boldsymbol{u}_V^{(n)})} = v^{(n)}\} \prod_{U \in \boldsymbol{U}} \theta_u. \tag{15}$$

2. *Sample $\boldsymbol{\xi}$.* Parameters $\boldsymbol{\xi}$ define deterministic causal mechanisms. For a given parameter $\xi_V^{(pa_V, \boldsymbol{u}_V)} \in \boldsymbol{\xi}$ its conditional distribution is given by $P(\xi_V^{(pa_V, \boldsymbol{u}_V)} = v \mid \bar{\boldsymbol{v}}, \bar{\boldsymbol{u}}) = 1$ if there exists a sample $(\boldsymbol{v}^{(n)}, pa_V^{(n)}, \boldsymbol{u}^{(n)})$ for some $n$, where $n$ iterates over the samples of $\boldsymbol{u}$ from step 1 and $\boldsymbol{v}$ associated with the *subset* of domains in which exogenous probabilities match the target domain, such that $\xi_V^{(pa_V^{(n)}, \boldsymbol{u}_V^{(n)})} = v^{(n)}$. Otherwise, $P(\xi_V^{(pa_V, \boldsymbol{u}_V)} = v \mid \bar{\boldsymbol{v}}, \bar{\boldsymbol{u}})$ is given by a uniform discrete distribution over its support $\text{supp}_V$.

3. *Sample $\boldsymbol{\theta}$*. Let $\boldsymbol{\theta}_U = (\theta_1, \ldots, \theta_{d_U}) \in \boldsymbol{\theta}$ be the parameters that define the probability vector of possible values of variables $U \in \boldsymbol{U}_C$. Its conditional distribution is given by,

$$\boldsymbol{\theta}_U \mid \bar{\boldsymbol{v}}, \bar{\boldsymbol{u}} \sim \texttt{Dirichlet}\left(\alpha_1 + \beta_1, \ldots, \alpha_{d_U} + \beta_{d_U}\right),$$

where $\beta_i := \sum_n \mathbb{1}\{u^{(n)} = u_i\}$. Similarly, $n$ iterates over the samples of $\boldsymbol{u}$ from step 1 associated with the subset of domains in which exogeneous probabilities match the target domain.

## A.2   Implementing Constraints

Iterating this procedure forms a Markov chain with the invariant distribution $P(\boldsymbol{u}, \boldsymbol{\xi}, \boldsymbol{\theta} \mid \bar{\boldsymbol{v}})$. This naturally enforces the soft constraint $P^{\mathcal{N}^i}(\boldsymbol{v}) = P^i(\boldsymbol{v}), i \in \{1, 2, \ldots, K, *\}$ for the CMs defined by the sampled parameters. The posterior distributions of the subset of $(\boldsymbol{\theta}^{\mathcal{N}^*}, \boldsymbol{\xi}^{\mathcal{N}^*})$ for which invariances across domains are assumed are then matched with the posterior distribution inferred from source data. The constraint $P^{\mathcal{N}^i}(r_V) = P^{\mathcal{N}^*}(r_V), i \in \{1, 2, \ldots, K, *\}, V \notin \Delta_{i,*}$ is enforced by generating $\theta_U^{\mathcal{N}^*}$ from the prior such that $P^{\mathcal{N}^*}(r_V) := \sum_{u \in \Omega} \theta_u^{\mathcal{N}^*} = \sum_{u \in \Omega} \theta_u^{\mathcal{N}^i} := P^{\mathcal{N}^i}(r_V), V \notin \Delta_{i,*}$ where $\Omega$ denotes the partition of $\text{supp}_U$ that is expressed by $R_V$.

The query is then approximated by plugging the $T$ MCMC samples into the query $\phi_{\mathcal{N}*}$ to obtain $\phi_{\mathcal{N}*}^{(1)}, \ldots, \phi_{\mathcal{N}*}^{(T)}$ and

$$\hat{q}_{\max} := \sup\{x : \sum_t \mathbb{1}\{\phi_{\mathcal{N}*}^{(t)} \leqslant x\} = \alpha\}. \tag{16}$$

for a chosen value of confidence value $\alpha$.

**Example 5** (Example 3 continued). Consider again the evaluation of the risk $R_{P*}(h) := P^{\mathcal{N}^*}(Y \neq h(X))$ given the classifier $h(x) = \neg x$. We are data sampled from $P^1(x, y), P^2(x, y)$. For every SCM $\mathcal{M}$, there exists an SCM of the described format specified with only a distribution $P(r_X, r_Y)$, where,

$$\text{supp}_{R_X} = \{0, 1\}, \quad \text{supp}_{R_Y} = \{y = 0, y = 1, y = x, y = \neg x\}. \tag{17}$$

Thus, the joint distribution $P(u_{XY}) = P(r_X, r_Y)$ can be parameterized by a vector in 8-dimensional simplex. The canonical SCMs associated with each of the SCMs $\mathcal{M}^1, \mathcal{M}^2, \mathcal{M}^*$, are denoted $\mathcal{N}^1, \mathcal{N}^2, \mathcal{N}^*$, for which $\boldsymbol{V} = \{X, Y\}, \boldsymbol{U} = \{U_{XY}\}$ and $\text{supp}_{U_{XY}} = \{1, \ldots, 8\}$. The partial task can be translated into an optimization problem aiming to to find the upper-bound for the risk $R_{P*}(h)$ for the classifier $h(x) = \neg x$:

$$\max_{\mathcal{N}^1, \mathcal{N}^2, \mathcal{N}*} P^{\mathcal{N}^*}(Y \neq \neg X) \tag{18}$$

$$\text{s.t. } P^{\mathcal{N}^1}(r_Y) = P^{\mathcal{N}^*}(r_Y), \quad P^{\mathcal{N}^2}(r_X) = P^{\mathcal{N}^*}(r_X) \qquad (Y \notin \Delta_1, \text{ and } X \notin \Delta_2)$$

$$P^{\mathcal{N}^1}(x, y) = P^1(x, y), \quad P^{\mathcal{N}^2}(x, y) = P^2(x, y) \qquad (\text{matching source dists})$$

With the Gibbs sampler outlined above, we obtain samples from the posterior distribution $P(\theta^{\mathcal{N}^1}, \theta^{\mathcal{N}^2}, \xi^{\mathcal{N}^1}, \xi^{\mathcal{N}^2} \mid \bar{\boldsymbol{v}})$. $\theta^{\mathcal{N}^1}, \theta^{\mathcal{N}^2}$ encode the probabilities $P^{\mathcal{N}^1}(U_{XY} = u), P^{\mathcal{N}^2}(U_{XY} = u)$ and are instantiated as two-dimensional arrays of shape $(2, 4)$ such that, e.g., $P^{\mathcal{N}^1}(r_Y) = \sum_{\text{dim. } 1} \theta^{\mathcal{N}^1}$, with $r_Y \in \{1, 2, 3, 4\}$ and similarly $P^{\mathcal{N}^1}(r_X) = \sum_{\text{dim. } 0} \theta^{\mathcal{N}^1}$, with $r_X \in \{1, 2\}$.

To enforce the constraints $P^{\mathcal{N}^1}(r_Y) = P^{\mathcal{N}^*}(r_Y), \quad P^{\mathcal{N}^2}(r_X) = P^{\mathcal{N}^*}(r_X)$ it thus suffices to sample $\theta^{\mathcal{N}^*}$ from the prior Dirichlet distribution (as it has not been updated with data) and re-scale the outcomes such that the partial row and column sums satisfy the corresponding partial row and column sums computed from the MCMC samples of $P(\theta^{\mathcal{N}^1}, \theta^{\mathcal{N}^2} \mid \bar{\boldsymbol{v}})$. The resulting MCMC parameters $(\boldsymbol{\theta}^{\mathcal{N}^*}, \boldsymbol{\xi}^{\mathcal{N}^*})$ are then valid samples from the posterior distribution $P(\theta^{\mathcal{N}^*}, \xi^{\mathcal{N}^*} \mid \bar{\boldsymbol{v}})$ subject to assumed constraints, and the risk could be computed by plugging those samples into $R_{P*}(h) := P^{\mathcal{N}^*}(Y \neq h(X))$ to obtain $R_{P*}(h)^{(1)}, \ldots, R_{P*}(h)^{(T)}$ and evaluating

$$\hat{q}_{\max} := \sup\{x : \sum_t \mathbb{1}\{R_{P*}(h)^{(t)} \leqslant x\} = \alpha\}. \tag{19}$$

for a chosen value of confidence value $\alpha$. □

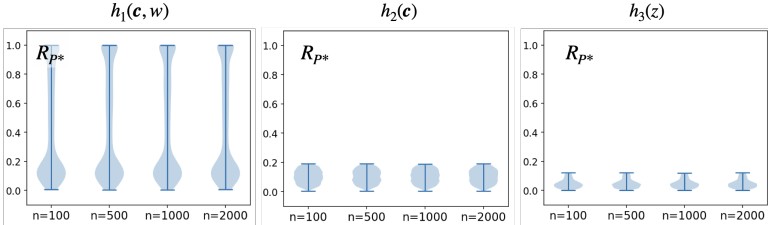

Figure 7: Violin plots that describe MCMC samples of $R_{P*}(h)$ for Example 2. The upper end-point is an estimate of $\max R_{P*}(h)$. $n$ stands for the number of source domain samples used as a conditioning set in the posterior evaluation.

The following Theorem shows that $\hat{q}_{max}$ converges to the true (tight) bounds $q_{max}$ for the unknown query $R_{P*}(h)$.

**Theorem 4.** $\hat{q}_{max}$ *defined in Eq.* (19) *is a valid upper bound on* $q_{max}$ *for any sample size, and coincides with* $q_{max}$ *as the sample size increases to infinity.*

*Proof.* Let $\Theta$ denote the collection of parameters $\xi, \theta$ of discrete SCMs that generate the observed data from $P^1, P^2, \ldots$. We assume that the prior distribution on $\xi, \theta$ has positive support over the domain of $\Theta$. That is, the probability density function $\rho(\xi) > 0$ and $\rho(\theta) > 0$ for every possible realization of $\xi, \theta$. By the definition of $\Theta$, for every pair of parameter $(\xi, \theta) \in \Theta$, it must be compatible with the dataset $\bar{v}$, i.e., $P(\bar{v} \mid \xi, \theta) > 0$. Similarly, given that the prior has positive support in $\Theta$, $P(\xi, \theta \mid \bar{v}) > 0$.

Note that parameters $(\xi, \theta) \in \Theta$ fully determine the optimal upper bound $q_{max}$ for $R_{P*}(h)$. And so this implies that $P(R_{P*}(h) < q_{max} \mid \bar{v}) > 0$, which by definition of a $100\%$ credible interval means that $R_{P*}(h) < \hat{q}_{max}$.

Next we show convergence of the posterior by way of convergence of the likelihood of the data given one SCM $\mathcal{M}$. For increasing sample size the posterior will, with increasing probability, be low for any parameter configuration, i.e. for any $(\xi, \theta) \notin \Theta$. By the definition of the optimal upper bound $q_{max}$ given by the solution to the partial identification task,

$$P(\bar{v} \mid R_{P*}(h) < q_{max}) \to_p 1. \tag{20}$$

Therefore if the prior on parameters $(\xi, \theta)$ defining SCMs is non-zero for any $\mathcal{M}$ compatible with the data and assumptions, also the posterior converges,

$$P(R_{P*}(h) < q_{max} \mid \bar{v}) \to_p 1, \tag{21}$$

which is the definition of the credible value $\hat{q}_{max}$ as the $100^{th}$ quantile of the posterior distribution, which coincides with $q_{max}$ asymptotically. $\qquad\square$

# B  Additional Experiments and Details

This section includes experimental details not covered in the main body of this paper as well as additional examples to illustrate our methods, including the Bayesian inference approach.

For the approximation of credible intervals and expectations required for the Bayesian inference approach, we draw 10,000 samples from posterior distributions $P(\cdot \mid \bar{v})$ after discarding $2,000$ samples as burn-in. The results will be given a violin plots that encode the full posterior distribution of the query of interest, here the target error $R_{P*}(h)$ of a classifier $h$. The worst-case target error can then be read as the upper end-point of the posterior distribution.

For completeness, we provide MCMC results for Examples 2 and 3, analyzed in the main body of this paper, in Figs. 7 and 8, respectively. One could check that the upper bounds match with the analysis in the main body of this paper.

## B.1  Additional Examples

This section adds additional synthetic examples to illustrate our methods.

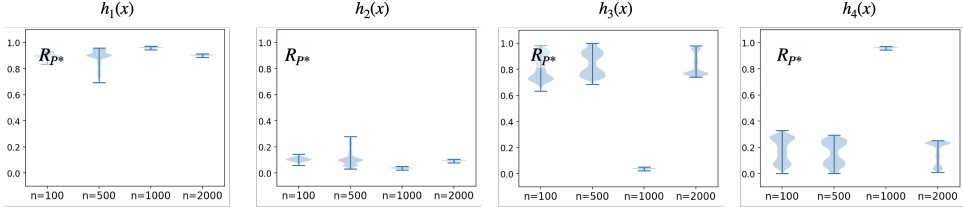

Figure 8: Violin plots that describe MCMC samples of $R_{P*}(h)$ for Example 3. The upper end-point is an estimate of $\max R_{P*}(h)$. $n$ stands for the number of source domain samples used as a conditioning set in the posterior evaluation. Recall that $h_1(x) := x, h_2(x) := \neg x, h_3(x) := 0, h_4(x) := 1$.

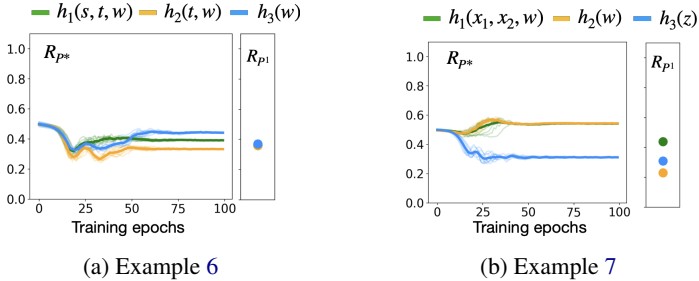

$\qquad$ (a) Example 6 $\qquad\qquad\qquad$ (b) Example 7

Figure 9: NCM experimental results on Examples 6 and 7.

**Example 6.** This experiment is inspired by the debate around the relationship between smoking and lung cancer in the 1950's [37], and the corresponding selection diagram is shown in Figure 12a. We consider $\mathbb{M} : \{\mathcal{M}^1, \mathcal{M}^*\}$ that describe the effect of an individual's smoking status $S$ on lung cancer $C$, including related measured variables such presence of tar in the lungs $T$, and demographic factors $W$. The data generating mechanism is given by

$$
\mathcal{M}^i = \begin{cases}
V & = \{W, S, T, C\} \\
U & = \{U_W, U_S, U_T, U_{SC}\} \\
\mathcal{F} & = \begin{cases}
f_W(u_W) & = u_W, \text{ for } W \in \boldsymbol{W}, U_W \in \boldsymbol{U}_W \\
f_S(\boldsymbol{w}, u_S, u_{SC}) & = \begin{cases} 1, & \text{if } \sum_i \frac{w_i}{d} + u_{SC} + 1.5 * u_s - 1 > 0 \text{ and } i = 1 \\ 1, & \text{if } \sum_i \frac{w_i}{d} + u_{SC} + u_S - 2 > 0 \text{ and } i = * \\ 0, & \text{otherwise} \end{cases} \\
f_T(s, u_T) & = \begin{cases} 1, & \text{if } s - 0.5u_T - 1 > 0 \\ 0, & \text{otherwise} \end{cases} \\
f_C(\boldsymbol{w}, u_C, u_{SC}) & = \begin{cases} 1, & \text{if } t - \sum_i \frac{w_i}{d} + u_{SC} - 1 > 0 \\ 0, & \text{otherwise} \end{cases}
\end{cases} \\
P(\boldsymbol{U}) & \text{defined such that } U_S, U_T, U_{SC} \sim Bern(0.5), U_W \sim N(0,1), W \in \boldsymbol{W},
\end{cases}
$$

Note that $\boldsymbol{\Delta} = \{S\}$ as the mechanism for $S$ differs across domains while the mechanisms for all other variables are assumed invariant. The quantity to upper-bound is the target mean squared error: $R_{P*}(h) := \mathbb{E}_{P*}\left[(C - h)^2\right]$ of cancer prediction algorithms $h \in \{h_1(w, s, t) = \mathbb{E}_{P^1}[C \mid w, s, t], h_2(w, t) = \mathbb{E}_{P^1}[C \mid w, t], h_3(w) = \mathbb{E}_{P^1}[C \mid w]\}$ given data from $P^1$ and $\mathcal{G}^{\boldsymbol{\Delta}}$.

The results for the NCM approach are given in Fig. 9a. We observe that despite the discrepancy in $S$, all methods maintain an error of close to 0.4.

The results for the Gibbs sampling approach are given in Fig. 10. The violin plots encode the full posterior distribution of the query of interest, here the target error $R_{P*}(h)$ of a classifier $h$. The worst-case target error can then be read as the upper end-point of the posterior distribution. We observe that the upper-bounds from the NCM and MCMC approach approximately match. □

**Example 7.** This experiment considers the design of prediction rules for the development of Alzheimer's disease in a target hospital $\mathcal{M}^*$ in which no data could be recorded, and the corresponding selection diagram is shown in Figure 12b. The observed variables are given by $\boldsymbol{V} = \{X_1, X_2, W, Y, Z\}$. Among those, $X_1$ and $X_2$ are treatments for hypertension and clinical depression, respectively, both known to influence Alzheimer's disease $Y$, and blood pressure $W$. $Z$ is a symptom of Alzheimer's. Their biological mechanisms are somewhat understood, e.g. the

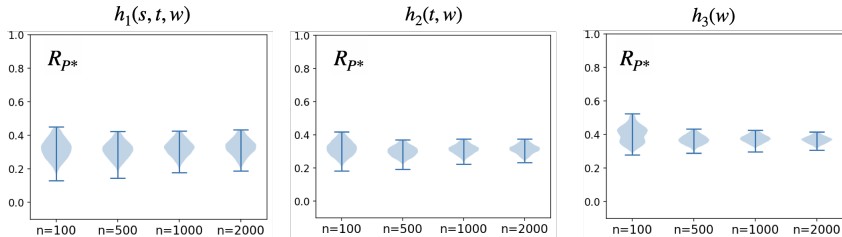

Figure 10: Violin plots that describe MCMC samples of $R_{P*}(h)$ for Example 6. The upper end-point is an estimate of $\max R_{P*}(h)$. $n$ stands for the number of source domain samples used as a conditioning set in the posterior evaluation.

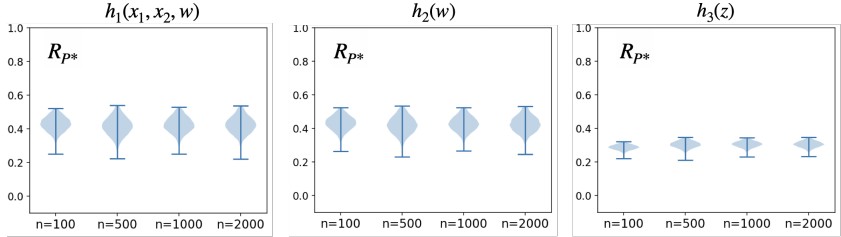

Figure 11: Violin plots with MCMC samples for Example 7. $n$ stands for the number of source domain samples used as a conditioning set in the posterior evaluation.

effect of hypertension is mediated by blood pressure $W$, although several unobserved factors, such as physical activity levels and diet patterns, are expected to simultaneously affect both conditions. We assume that hypertension and clinical depression are not known to affect each other, although it's common for patients with clinical depression to simultaneously be at risk of hypertension (expressed through the presence of an unobserved common cause). More specifically, investigators have access to data from a related study conducted in domain $\mathcal{M}^1$. SCMs $\mathbb{M} : \{\mathcal{M}^1, \mathcal{M}^*\}$ are given as follows,

$$\mathcal{M}^i = \begin{cases} \boldsymbol{V} & = \{X_1, X_2, W, Y, Z\} \\ \boldsymbol{U} & = \{U_{WY}, U_{X_2}, U_W, U_{X_1 X_2}, U_Z\} \\ \mathcal{F} & = \begin{cases} f_{X_1}(U_{X_1 X_2}) & = \begin{cases} 1, & \text{if } U_{X_1 X_2} > 0 \\ 0, & \text{otherwise} \end{cases} \\ f_{X_2}(U_{X_1 X_2}, U_{X_2}) & = \begin{cases} 1, & \text{if } U_{X_1 X_2} + U_{X_2} > 0 \\ 0, & \text{otherwise} \end{cases} \\ f_W(X_1, U_{WY}, U_W) & = \begin{cases} 1, & \text{if } X_1 + U_{WY} + 1.5 U_W - 1 > 0 \text{ and } i = * \\ 1, & \text{if } X_1 + U_{WY} - U_W + 1 > 0 \text{ and } i = 1 \\ 0, & \text{otherwise} \end{cases} \\ f_Y(W, X_1, U_{WY}) & = \begin{cases} 1, & \text{if } W - U_{WY} + 0.1 X_1 - 1 > 0 \\ 0, & \text{otherwise} \end{cases} \\ f_Z(Y, U_Z) & = \begin{cases} 1, & \text{if } Y + U_Z > 0.5 \\ 0, & \text{otherwise} \end{cases} \end{cases} \\ P(\boldsymbol{U}) & \text{defined such that } U_{WY}, U_{X_2}, U_W, U_{X_1 X_2}, U_Z \sim \mathcal{N}(0, 1), \end{cases}$$

Note that $\boldsymbol{\Delta} = \{W\}$ as the mechanism for $W$ differs across domains while the mechanisms for all other variables are assumed invariant. In this example, we aim at upper-bounding the target mean squared error: $R_{P*}(h) := \mathbb{E}_{P*}[(C - h)^2]$ of cancer prediction algorithms $h \in \{h_1(x_1, x_2, w) = \mathbb{E}_{P^1}[Y \mid x_1, x_2, w], h_2(w) = \mathbb{E}_{P^1}[Y \mid w], h_3(z, t) = \mathbb{E}_{P^1}[Y \mid z]\}$ given data from $P^1$ and $\mathcal{G}^{\Delta*1}$.

The results for the NCM approach are given in Fig. 9b. We observe that the discrepancy in $W$ leads to poor performance for all methods (chance level) except for $h_3$ that outperforms.

The results for the Gibbs sampling approach are given in Fig. 11. The violin plots encode the full posterior distribution of the query of interest, here the target error $R_{P*}(h)$ of a classifier $h$. The worst-case target error can then be read as the upper end-point of the posterior distribution. We observe that the upper-bounds from the NCM and MCMC approach approximately match. $\qquad\square$

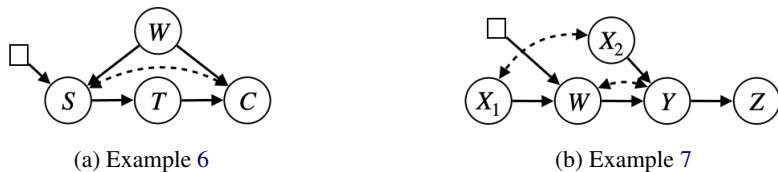

|               |               |
| :-----------: | :-----------: |
| (a) Example 6 | (b) Example 7 |

Figure 12: Selection diagrams for additional experiments

## B.2 More on Colored MNIST

Consider handwritten grayscale digits $W \in [0,1]^{28 \times 28}$ that are annotated with $Y \in \{0, 1, \ldots, 9\}$ and colored with $C \in \{\text{red}, \text{green}\}$, resulting in colored images $Z \in [0,1]^{28 \times 28 \times 3}$. What follows describes the underlying SCM for domain $i \in \{1, 2, *\}$:

$$
\mathcal{M}^i : \begin{cases} W, U_Y, U_C, U_Z \sim P(w) \cdot P(u_Y) \cdot P(u_C) \cdot P(u_Z) \\ \mathcal{F}^i : \begin{cases} Y \leftarrow f_Y(W, U_Y) & \text{(The annotation mechanism)} \\ C \leftarrow f_C^i(Y, U_C) & \text{(The choice of color based on digit)} \\ Z \leftarrow W \cdot C^\top + U_Z & \text{(Coloring image } W \text{ with color } C) \end{cases} \end{cases}
$$

In words, the grayscale image of handwritten digits $W$ is generated according to a distribution $P(w)$ shared across all domains. The label $Y$ is the annotation of the image with the corresponding digit through mechanism $f_Y$ shared across all domains; the variable $U_Y$ accounts for the possible error in annotation. Next, the color is chosen based on the digit $Y$ following some stochastic policy $f_C^i(\cdot, U_C)$ that changes across the source and target domains. Finally, the colored image $Z$ is produced by product of the grayscale image $W$ and the color $C$; exogenous variable $U_Z$ accounts for possible noise in coloring.

We have a classifier $h : \Omega_Z \rightarrow \Omega_Y$ at hand, and the task is to assess its generalizability. Consider the following derivation:

$$
P^*(z, y) = \sum_c P^*(y, c, z) \tag{22}
$$

$$
= \sum_c P^*(y) \cdot P^*(c \mid y) \cdot P^*(z \mid c, y) \tag{23}
$$

$$
= P^*(y) \sum_c P^*(c \mid y) \cdot P^{1,2}(z \mid c, y) \qquad S_1, S_2 \perp\!\!\!\perp_d Z \mid C, Y \tag{24}
$$

$$
= P^{1,2}(y) \sum_c P^*(c \mid y) \cdot P^{1,2}(z \mid c, y) \qquad S_1, S_2 \perp\!\!\!\perp_d Y \tag{25}
$$

Motivated by the above derivation, we use the source data drawn from $P^1, P^2$ and train the generative models $P(y; \eta_Y), P(z \mid y, c; \eta_Z)$ to approximate sampling from the distributions $P^{1,2}(y), P^{1,2}(z \mid y, c)$, respectively. The former generates a random digit $Y$ according to the distribution of label in the source domain, and the latter generates a colored picture $Z$ by taking color $C$ and digit $Y$ as the input. Also, we use an NCM with parameter $\theta_C^*$ to model the c-factor $P^*(c \mid do(y)) = P^*(c \mid y)$. We can now rewrite the risk as follows:

$$
R_{P*}(h) = \sum_{z,y} |y - h(z)| \cdot P^{1,2}(y) \sum_c P^*(c \mid y) \cdot P^{1,2}(z \mid c, y) \tag{26}
$$

$$
= \mathbb{E}_{Y \sim P(y; \eta_Y)} \Big[ \sum_c P(c \mid y; \theta_C^*) \cdot \mathbb{E}_{Z \sim P(z \mid c, y; \eta_Z)} [|Y - h(Z)|] \Big]. \tag{27}
$$

By maximizing the above w.r.t. the free parameter $\theta_C^*$, we achieve the worst-case risk of the classifier.

## B.3 Reproducibility

For the synthetic experiments, we used feed-forward neural networks with 7 layers and $128 \times 128$ neurons in each layer. The activation for all layers is ReLu, but for the last layer which is a sigmoid since $f_{\theta_V}$ outputs the probability of $V = 1$. For evaluation, at each epoch, we used 1000 samples from the joint distribution. The data generative process for all experiments is provided in the corresponding

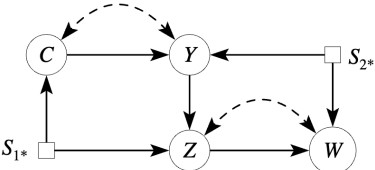

Figure 13: Selection diagram of Example 8

example. We used Adam optimizer for training the Neural networks. In CMNIST example, we used a standard implementation of a conditional GAN [23] trained over 200 epochs with a batch-size of 64. The learning rate of Adam was set to 0.0002. The architecture of the generator is given by a 5 layer feed-forward neural network with Batch normalization and Leaky-ReLu activations.

## C  Extended Discussion on Algorithms

In this section, we elaborate more on the algorithms presented in the paper.

### C.1  Examples of Neural-TR (Algorithm 1)

In the next examples, we follow Algorithm 1 to compute the worst-case risk of a classifier.

**Example 8** (Simplify). Consider a system of SCMs $\mathcal{M}^1, \mathcal{M}^2 \mathcal{M}^*$ over $\boldsymbol{X} = \{C, Z, W\}$ and $Y$ that induces the selection diagram shown in Figure 13. Suppose we would like to assess the risk of a classifier $h(z)$. Following Theorem 2, the naive approach requires us to parameterize three NCMs $\theta^1, \theta^2, \theta^*$ over the variables $\boldsymbol{X}, Y$, and then proceed with the maximization of the target quantity $R_{P\mathcal{N}*}(h) = \mathbb{E}_{P*}[\mathbb{1}\{Y = h(Z)\}]$. Notably, the latter depends only on $P^*(y, z)$. We can rewrite the risk of $h$ as follows:

$$\mathbb{E}_{Y,Z \sim P*(y,z)}[\mathbb{1}\{Y \neq h(Z)\}] = \sum_{z,y} \mathbb{1}\{y \neq h(z)\} \cdot P^*(y, z) \tag{28}$$

$$= \sum_{z,y} \mathbb{1}\{y \neq h(z)\} \cdot P^*(y) \cdot P^*(z \mid y) \qquad \text{(factorization)} \tag{29}$$

$$= \sum_{z,y} \mathbb{1}\{y \neq h(z)\} \cdot P^*(y) \cdot P^2(z \mid y) \qquad (Y \mid Z \perp\!\!\!\perp_d S_2) \tag{30}$$

$$= \sum_{z,y} \mathbb{1}\{y \neq h(z)\} \cdot P^2(z \mid y) \cdot \sum_c P^*(y, c) \tag{31}$$

This new expression for the objective function depends only on the unknown $P^*(y, c)$, a so-called ancestral c-factor, that can generally be expressed as $P^*(\boldsymbol{a} \mid do(pa_{\boldsymbol{A}})), \boldsymbol{A} = \{C, Y\}$. In the following, we argue that to partially transport the risk we only need to parameterize the SCMs over ancestral c-factors that are not transportable. Specifically, the partial transportation problem can be restated as follows:

$$\max_{\theta^1, \theta^2, \theta^*} \quad \mathbb{E}_{\boldsymbol{U}_{CY}}\Big[\sum_{y,c,z} P(y, c \mid \boldsymbol{U}_{CY}; \theta^*) \cdot P(z \mid y; \eta^2) \cdot \mathbb{1}\{y \neq h(z)\}\Big] \tag{32}$$

$$+ \Lambda \cdot \Big( \sum_{y,c \in D^1} \mathbb{E}_{\boldsymbol{U}_{CY}}[\log P(y, c \mid U_{CY}; \theta^1)] + \sum_{y,c \in D^2} \mathbb{E}_{\boldsymbol{U}_{CY}}[\log P(y, c \mid U_{CY}; \theta^2)]\Big)$$

$$\text{s.t. } \theta^*[C] = \theta^2[C], \quad \theta^*[Y] = \theta^1[Y].$$

In the above, $D^i \sim P^i(c, y, z, w)$ denotes the source data, and $P(z \mid y; \eta^2)$ is a probabilistic model of $P^2(z \mid y)$ learned using the data $D^2$. □

**Example 9** (Partial-TR illustrated). Consider a system of SCMs $\mathcal{M}^1, \mathcal{M}^2, \mathcal{M}^*$ over the binary variables $\boldsymbol{X} = \{X_1, X_2, \dots, X_9\}$ and $Y$ that induces the selection diagram shown in Figure 14. Consider the classifier $h(x_1, x_4) = x_1 \lor x_4$. The objective is partial transportation of the risk of $h$, expressed as follows:

$$R_{P*}(h) = P^*(Y \neq h(X_1, X_4)) \tag{33}$$

$$= \mathbb{E}_{X_1, X_4, Y \sim P*}[\mathbb{1}\{Y \neq X_1 \lor X_4\}]. \tag{34}$$

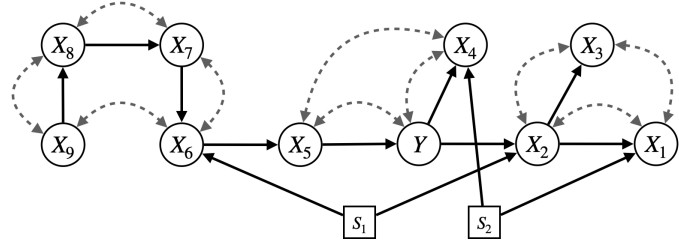

Figure 14: Selection diagram of Example 9

The latter indicates that $\psi(X_1, X_4, Y) := \mathbb{1}\{Y \neq X_1 \vee X_4\}$ must be passed to the algorithm. The objective function is then expressed as:

$$R_{P*}(h) = \mathbb{E}_{P*}[\psi(X_1, X_4, Y)] = \sum_{x_1, x_4, y} \mathbb{1}\{Y \neq X_1 \vee X_4\} \cdot P^*(x_1, x_4, y). \tag{35}$$

Next, we focus on transporting $P^*(x_1, x_4, y)$. First, we compute the ancestral set using the selection diagram;

$$\boldsymbol{A} = An(X_1, X_4, Y) = \{X_1, X_2, X_4, Y, X_5, X_6, X_7, X_8, X_9\}. \tag{36}$$

and we decompose this set into c-components:

$$\boldsymbol{A}_1 = \{X_1, X_2\}, \quad \boldsymbol{A}_2 = \{X_4, Y, X_5\}, \quad \boldsymbol{A}_3 = \{X_6, X_7, X_8, X_9\}. \tag{37}$$

Next, we form the expression below:

$$P^*(x_1, x_4, y) := \sum_{x_2, x_5, \dots, x_9} P^*(x_1, x_2 \mid do(y)) \cdot P^*(y, x_4, x_5 \mid do(x_6)) \cdot P^*(x_6, \dots, x_9). \tag{38}$$

Notice,

$$P^*(\boldsymbol{a}_2 \mid do(x_6)) \stackrel{\text{rule 2 do-calc.}}{=} P^*(\boldsymbol{a}_2 \mid x_6) \stackrel{S_1 \perp\!\!\!\perp_d Y, X_4, X_5 \mid X_6}{=} P^1(\boldsymbol{a}_2 \mid x_6), \tag{39}$$

$$P^*(\boldsymbol{a}_3) \stackrel{S_2 \perp\!\!\!\perp_d \boldsymbol{A}_3 \mid X_6}{=} P^2(\boldsymbol{a}_3). \tag{40}$$

Thus, we use the source data $D^1, D^2$ to learn the generative model $P(\boldsymbol{a}_2 \mid x_6; \eta^1_{\boldsymbol{A}_2}), P(\boldsymbol{a}_3; \eta^2_{\boldsymbol{A}_3})$ to approximate sampling from $P^1(\boldsymbol{a}_2 \mid x_6), P^2(\boldsymbol{a}_3)$ respectively. We plug these models as constants into Eq. 35.

Since $S_{*1}, S_{*2}$ are pointing to the variables $X_2, X_1$, respectively, the first term $P^*(x_1, x_2) \mid do(y))$ in Eq. 38 can not be directly transported from neither of the source domains. Thus, we need to parameterize this c-factor using NCMs across all domains. We require the following properties:

1. **Parameter sharing**: Since $X_4, Y, X_5, X_7, X_8, X_9$ are not pointed by $S_1$, we share their mechanisms across all domains. Also, since $X_2, X_6$ are not pointed by $S_2$, we set $\theta^*_{\{X_2, X_6\}} = \theta^2_{\{X_2, X_6\}}$. These constraints are stored in $\mathbb{C}_{\text{expert}}$ in the Algorithm.

2. **Source data**: To enforce $\theta^1, \theta^2$ to be compatible with the source data $D^1, D^2$, we compute the likelihood of the data w.r.t. the parameters, as follows:

$$\mathcal{L}_{\text{likelihood}} := \sum_{i=1}^{2} \Big( \sum_{\langle x_1, x_2, y \rangle \in D^i} \mathbb{E}_{\boldsymbol{U}_{X_1, X_2}}[\log P(x_1, x_2 \mid y, \boldsymbol{U}_{X_1, X_2}; \theta^i_{X_2})] \tag{41}$$

We plug $P(x_1, x_2 \mid do(y); \theta^*_{\boldsymbol{A}_1})$ into Eq. 38. Finally, we use stochastic gradient ascent to maximize the objective function in Eq. 35 regularized by an additive term $\Lambda \cdot \mathcal{L}_{\text{likelihood}}$ that encourages the likelihood of the data w.r.t. the parameters of the source NCMs. $\square$

### C.2 Illustration of CRO (Algorithm 2)

First, we initialize with a random classifier. One may also warm start with a reasonable guess such as empirical risk minimizer defined as,

$$h_{\text{ERM}} \in \underset{h:\Omega_{\boldsymbol{X}}\to\Omega_{\boldsymbol{Y}}}{\arg\min} \sum_{i=1}^{K} \sum_{\boldsymbol{x},y\in D^i} \mathcal{L}(y, h(\boldsymbol{x})). \tag{42}$$

Throughout the runtime of the algorithm we accumulate instances of distributions that we obtain via Neural-TR (Alg. 1). At each step, these distribution are aimed to maximize the risk of the classifier at hand. In this sense, Neural-TR can be viewed as an adversary, and the CRO can be viewed as a game between two players:

1. **Neural-TR.** Searches over the spaces of plausible target domains that are characterized by the source data and the domain relatedness encoded in the selection diagram, to find a distribution that is hard to generalize to using the classifier at hand.

2. **group DRO [32]** Updates the classifier at hand by minimizing the maximum risk over the distributions produced by Neural-TR so far, that is,

$$\min_{h:\Omega_{\boldsymbol{X}}\to\Omega_{\boldsymbol{Y}}} \max_{D\in\mathbb{D}^*} \frac{1}{|D|} \cdot \sum_{\boldsymbol{x},y\in D} \mathcal{L}(y, h(\boldsymbol{x})). \tag{43}$$

For more information about group DRO, see Appendix D.2.

The equilibrium of the above happens if the worst-case risk obtained by Neural-TR almost coincides with the risk obtained by group DRO, i.e.,

$$R_{P(\boldsymbol{x},y;\hat{\theta})}(h) - \max_{D\in\mathbb{D}^*} \frac{1}{|D|} \cdot \sum_{\boldsymbol{x},y\in D} \mathcal{L}(y, h(\boldsymbol{x})) < \delta. \tag{44}$$

Once this is achieved, we stop the search and return the classifier at hand. When the game is not at equilibrium, we would have a difference larger than $\delta$, meaning that the new target domain $\hat{\theta}^*$ has enough novelty to forces the classifier at hand to perform at least $\delta$ worse than what it achieves over the existing distributions in $\mathbb{D}^*$. Therefore, we draw samples $D^* \sim P(\boldsymbol{x}, y; \hat{\theta})$ and add them to our collection $\mathbb{D}^*$. As shown in Theorem 3 this game reaches the equilibrium in finitely many steps, and the classifier that we return has the best worst-case risk w.r.t. the selection diagram $\mathcal{G}^{\boldsymbol{\Delta}}$ and the source distributions $\mathbb{P}$. The conceptual Figure 15 shows the process of convergence of CRO.

It is important to note that although we employ group DRO as a subroutine in our CRO algorithm, we do not use the source distributions directly. Instead, we use group DRO on the distributions obtained from Neural-TR. Note that under the assumptions encoded in the selection diagram, the target distribution distribution may be geometrically unrelated to the source distributions; the reason is that mechanistic relatedness of the target domain to the source domains (as indicated by the graph) do not translate directly to closeness of the entailed distributions under known distributional distance measures.

## D  Extended Related Work

In this section, we discuss some learning schemes based on invariant and robust learning that are proposed for domain generalization, including IRM and group DRO that are discussed in the experiments.

### D.1  Invariant Learning for Domain Generalization

Several common invariance criteria are extensively studied in the literature and proposed for the domain generalization task. A prominent idea is label conditional distribution invariance that seeks a representation $\phi$ such that $P^i(Y \mid \phi(\boldsymbol{X}))$ is equal across the source domains [29, 1, 13, 22]. These notions do not explicitly rely on an underlying structural causal model (SCM), although invariances are often justified by an underlying causal model [27, 2, 39, 31, 33]. Jalaldoust & Bareinboim

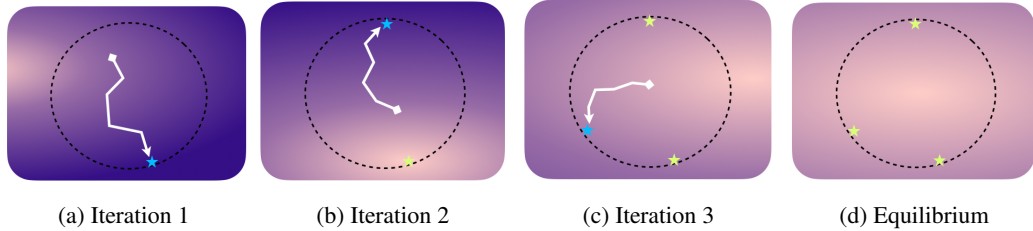

| (a) Iteration 1 | (b) Iteration 2 | (c) Iteration 3 | (d) Equilibrium |

Figure 15: Conceptual illustration of CRO. The rectangle represents the space of all distributions over $\boldsymbol{X}, Y$, and the circle inside it represents the subset of that are plausible target distributions, as characterized by the source distributions and selection diagram. Iteration 1: At first we start with some classifier that may or may not perform well for all distributions in the plausible subset; the darker spots indicate distributions that yield higher risk for the classifier at hand. Neural-TR uses gradient ascend steps to find an SCM that entails a distribution which yields the highest risk for the classifier at hand, i.e., the darkest spot within the plausible subset (likely at the boundary of it), shown by the star blue in Fig. (a). We register this distribution by taking samples from it and adding them to the collection $\mathbb{D}^*$. Iteration 2: We update the classifier at hand to have group robustness to the collection of distributions $\mathbb{D}^*$; in this case, only risk minimizer, since there is only one distribution in the collection. Now the distributions that are *close* to the registered distribution would entail small risk, thus, the region around the first star is now brighter. Once again, using Neural-TR we find a distribution that yields high risk for the classifier at hand. Iteration 3: We update the classifier, this time to minimize the risk on both registered distributions indicated with yellow starts using group DRO. Now the risk is smaller in most parts of the plausible set, though Neural-TR still finds another distribution at the boundary with high risk. Equilibrium: We update the classifier using group DRO over the three registered distributions. This time, the registered distributions correctly represent the plausible set, meaning that the maximum risk inside the plausible set is not significantly larger than what is achieved at the registered points through group DRO.

[17] studied the implicit assumptions that license generalizability of representations that satisfy the probabilistic relation $P^i(Y \mid \phi(\boldsymbol{X}))$. Although searching for such representation is practically challenging and in cases theoretically intractable. Thus, one may resort to achieving an approximate notion that serve as a proxy to invariance of $P^i(Y \mid \phi(\boldsymbol{X}))$; A well-known instance of such effort is invariant risk minimization [2], discussed below.

The paper [2] studies a constrained optimization problem called invariant risk minimization (IRM) in the context of domain generalization. In the notation of our paper, the IRM problem can be written as follows:

$$
\begin{aligned}
\min_{\phi, h} \quad & \sum_{i=1}^{K} \mathbb{E}_{P^i}[Y \neq h \circ \phi(\boldsymbol{X})] \\
\text{s.t.} \quad & h \in \operatorname*{arg\,min}_{\tilde{h}:\Omega_{\boldsymbol{R}} \to \{0,1\}} \mathbb{E}_{P^i}[Y \neq \tilde{h} \circ \phi(\boldsymbol{X})] \quad \forall i,
\end{aligned}
\tag{45}
$$

Where $\phi : \Omega_{\boldsymbol{X}} \to \Omega_{\boldsymbol{R}}$ is a representation, and $h : \Omega_{\boldsymbol{R}} \to \{0, 1\}$ is a classifier defined based on it. In words, a pair $h, \phi$ satisfies the invariant risk minimization property if $h \circ \phi$ attains the minimum risk across all classifiers defined based on $\phi$, across all source domains. The search procedure suggests choosing the classifier that satisfies the mentioned constraint, and achieves minimum risk on the pooled source data. The constrained optimization program above is highly non-convex and hard to solve in practice. To approximate the solution, the paper considers the Langrangian form below:

$$
h_{\text{IRM}} \in \min_{h_\theta:\Omega_{\boldsymbol{X}} \to \{0,1\}} \sum_{i=1}^{K} \mathbb{E}_{P^i}[Y \neq h_\theta(\boldsymbol{X})] + \lambda \cdot \|\nabla_\theta \mathbb{E}_{P^i}[Y \neq h_\theta(\boldsymbol{X})]\|^2.
\tag{46}
$$

In this program, $\theta$ parametrizes the classifier $h$, and the penalty term $\lambda$ accounts for how restrictive one wants to enforce the IRM constraint. In the extreme $\lambda = 0$ the objective equates to the vanilla ERM with all data pooled; on the other extreme, for $\lambda \to \infty$ ascertains that the solution is guaranteed to satisfy the IRM constraint.

Consider a representation that satisfies the original IRM constraint in Eq. 45. The optimal classifier defined over this representation is the bayes classifier, that uses $\frac{1}{2}$ level set of $P^i(Y = 1 \mid \phi(\boldsymbol{X}))$ as the decision boundary. This means that satisfying the IRM constraint implies a match between $\frac{1}{2}$ level-sets of $P^i(Y = 1 \mid \phi(\boldsymbol{X}))$ across all source domains. On the other hand, invariance of

$P^i(Y \mid \phi(\boldsymbol{X}))$ requires coincidence of every level-set across the source domains, and in this sense, the IRM constraint can be viewed as a proxy to the invariance property of $P^i(Y \mid \phi(\boldsymbol{X}))$. One can speculate that since IRM yields a proxy to invariance of $P^i(Y \mid \phi(\boldsymbol{X}))$, it might still exhibit generalization, though slightly weaker than what is derived from invariance of $P^i(Y \mid \phi(\boldsymbol{X}))$. However, IRM is shown to have poor domain generalizability, both theoretically (e.g., [30]) and empirically (e.g., [15]). Still, due to popularity of this method in the literature, we find it insightful to use the Neural-TR algorithm to find out what would be the worst-case risk of IRM. As shown in Fig. 4c, the worst-case performance of IRM is much worse than what is reported by [2] and [15]; the reason is that Neural-TR does not commit to one held-out domain, and instead it constructs an SCMs that is tailored to yield the poorest performance subject to the graph and source distributions.

### D.2  Group Robustness for Domain Generalization

Group Distributionally robust optimization (group DRO) [32] has been employed in the broad context of learning under uncertainty. In group DRO one seeks a single classifier that minimizes the risk on multiple distributions simultaneously. More specifically, the objective is minimizing the maximum risk among the source distributions, i.e.,

$$h_{\mathrm{DRO}} \in \arg\min \max_{i \in \{1,2,\dots,K\}} R_{P^i}(h) \tag{47}$$

This approach ensures that the learned classifier is optimal w.r.t. an unknown target domain that lies in the convex hull of the source distributions. In this sense, group DRO objective interpolates the perturbations that are represented in the source data to define an uncertainty set for the target distribution. On the other hand, in invariant learning the objective is to extrapolate the perturbations that are observed among the source domain by learning a representation that shields the label from these changes. In particular, [18, 31, 33] highlight the invariant-robust spectrum, and propose methods that have a free parameter which allows interpolating the two. In our experiments, we considered group DRO as a representative of methods in this category, and evaluated its worst-case performance in the Colored MNIST task, as shown in Figure 4c. Once again, we emphasize that this worst-case risk is much larger than what is shown in the benchmarks, e.g., by [15]. The reason is that the worst-case performance is obtained by Neural-TR that operates as an adversary, seeking a plausible target domain that is hardest to generalize to, subject to the assumptions encoded in the graph and the source data.

## E  Proofs

**Proof of Theorem 1**

Our results rely on the expressiveness of discrete SCMs, i.e. defined over variables $\{\boldsymbol{V}, \boldsymbol{U}\}$ with finite cardinalities. Discrete SCMs, introduced first in [3] and then in [42] have been shown to be "canonical" in the sense that they could represent all counterfactual distributions entailed by any SCM with the same induced causal diagram defined over finite $\boldsymbol{V}$. The following example illustrates this observation.

**Example 10** (The double bow). Let $\{X, Y, Z\}$ be binary variables. Consider two source domains defined based on the following SCMs:

$$\mathcal{M}^1 : \begin{cases} P^1(\boldsymbol{U}) : \begin{cases} U_X \sim \mathrm{Normal}(0,1) \\ U_{XY} \sim \mathrm{Normal}(0,1) \\ U_{ZY} \sim \mathrm{Normal}(0,1) \end{cases} \\ \mathcal{F}^1 : \begin{cases} X \leftarrow \mathbb{1}\{U_X + U_{XY} > 0\} \\ Y \leftarrow \mathbb{1}\{X - U_{XY} > 0\} \\ Z \leftarrow \mathbb{1}\{Y \cdot U_{ZY} > 0\} \end{cases} \end{cases} \qquad \mathcal{M}^* : \begin{cases} P^*(\boldsymbol{U}) : \begin{cases} U_X \sim \mathrm{Normal}(0,1) \\ U_{XY} \sim \mathrm{Normal}(0,1) \\ U_{ZY} \sim \mathrm{Normal}(0,1) \end{cases} \\ \mathcal{F}^* : \begin{cases} X \leftarrow \mathbb{1}\{U_X + U_{XY} > 0\} \\ Y \leftarrow \mathbb{1}\{-U_{XY} + 0.5 > 0\} \\ Z \leftarrow \mathbb{1}\{Y \cdot U_{ZY} > 0\} \end{cases} \end{cases}$$

The SCM $M^1$ induces a counterfactual probabilities, e.g. $P^{M^1}(x, y_x, z_y)$ for outcomes $x, y_x, z_y \in \{0,1\}$. [3] observed that such probabilities, defined over a finite set of events, may be generated with an equivalent model with a potentially large but finite set of discrete exogenous variables. [3] derived a canonical parameterization for the SCMs that induces the same graph but instead involves possibly correlated discrete latent variables $R_X, R_Y, R_Z$, where $R_X$ determines the functional that decides

$X$, $R_Y$ determines the functional that decides $Y$ based on $X$, and $R_Z$ determines the functional that decides $Z$ based on $Y$. [3] showed that for every SCM $\mathcal{M}$ with the same induced graph as $\mathcal{M}^1$ there exists an SCM of the described format specified with only a distribution $P(r_X, r_Y, r_Z)$, where,

$$\text{supp}_{R_X} = \{0, 1\},$$
$$\text{supp}_{R_Y} = \{y = 0, y = 1, y = x, y = \neg x\},$$
$$\text{supp}_{R_Z} = \{z = 0, z = 1, z = y, z = \neg y\}.$$

Thus, the joint distribution $P(r_X, r_Y, r_Z)$ can be parameterized by an 32-dimensional vector. □

This example illustrates a more general procedure, in which probabilities induced by an SCM over discrete endogenous variables $\boldsymbol{V}$ may be generated by a canonical model. This is formalized in the following lemma.

**Definition 7** (Canonical SCM). A canonical SCM is an SCM $\mathcal{N} = \langle \boldsymbol{U}, \boldsymbol{V}, \mathcal{F}, P(\boldsymbol{U}) \rangle$ defined as follows. The set of endogenous variables $\boldsymbol{V}$ is discrete. The set of exogenous variables $\boldsymbol{U} = \{R_V : V \in \boldsymbol{V}\}$, where $\text{supp}_{R_V} = \{1, \ldots, m_V\}$ (where $m_V = |\{h_V : \text{supp}_{pa_V} \to \text{supp}_V\}|$) for each $V \in \boldsymbol{V}$. For each $V \in \boldsymbol{V}$, $f_V \in \mathcal{F}$ is defined as $f_V(pa_V, r_V) = h_V^{(r_V)}(pa_V)$.

The following lemma establishes the expressiveness of canonical SCMs.

**Lemma 1** (Thm. 2.4 [42]). *For an arbitrary SCM $M = \langle \boldsymbol{U}, \boldsymbol{V}, \mathcal{F}, P(\boldsymbol{U}) \rangle$, there exists a canonical SCM $\mathcal{N}$ such that 1. $M$ and $\mathcal{N}$ are associated with the same causal diagram, i.e., $\mathcal{G}_M = \mathcal{G}_\mathcal{N}$. 2. For any set of counterfactual variables $\boldsymbol{Y_x}, \ldots, \boldsymbol{Z_w}$, $P^M(\boldsymbol{Y_x}, \ldots, \boldsymbol{Z_w}) = P^\mathcal{N}(\boldsymbol{Y_x}, \ldots, \boldsymbol{Z_w})$.*

In words, finite exogenous domains in canonical SCMs are sufficient for capturing all the uncertainties and randomness introduced by the (potentially) continuous latent variables in SCMs. Our goal will be to adapt the canonical parameterization of SCMs such that they entail the equality constraints specified by $\mathcal{G}^{\boldsymbol{\Delta}}$. The next example illustrates the implication of the constraints induced by $\mathcal{G}^{\boldsymbol{\Delta}}$ on the construction of canonical SCMs.

**Example 11** (Example 10 continued.). Consider $\mathcal{M}^1$ and $\mathcal{M}^*$ given in Example 10. The domain discrepancy set $\boldsymbol{\Delta}$ indicates that certain causal mechanisms need to match across pairs of the SCMs. For example, $\Delta_{1*} = \{Y\}$, which does not contain $\{X, Z\}$, and this implies that the functions $f_X, f_Z$ are invariant across $\mathcal{M}^1, \mathcal{M}^*$, and that the distribution of unobserved variables that are arguments of $f_Y, f_Z$, namely, $\{U_X, U_{XY}, U_{YZ}\}$ are invariant across $\mathcal{M}^1, \mathcal{M}^*$. The canonical parameterization of $\mathcal{M}^1$ is given by

$$\mathcal{N}^1 = \begin{cases} \boldsymbol{V} & = \{X, Y\} \\ \boldsymbol{U} & = \{R_X, R_Y, R_Z\} \\ \mathcal{F}^1 & = \begin{cases} f_X^1 : \text{supp}_{R_X} \to \text{supp}_X \\ f_Y^1 : \text{supp}_{R_Y} \times \text{supp}_X \to \text{supp}_Y \\ f_Z^1 : \text{supp}_{R_Z} \times \text{supp}_Y \to \text{supp}_Z \end{cases} \\ P^1(\boldsymbol{U}) & = P^1(R_X, R_Y, R_Z) \end{cases}$$

Analogously, the canonical parameterization of $\mathcal{M}^*$ is given by

$$\mathcal{N}^1 = \begin{cases} \boldsymbol{V} & = \{X, Y\} \\ \boldsymbol{U} & = \{R_X, R_Y, R_Z\} \\ \mathcal{F}^* & = \begin{cases} f_X^* : \text{supp}_{R_X} \to \text{supp}_X \\ f_Y^* : \text{supp}_{R_Y} \times \text{supp}_X \to \text{supp}_Y \\ f_Z^* : \text{supp}_{R_Z} \times \text{supp}_Y \to \text{supp}_Z \end{cases} \\ P^*(\boldsymbol{U}) & = P^*(R_X, R_Y, R_Z) \end{cases}$$

With these definitions, the restrictions in $\boldsymbol{\Delta}_{1*}$ impose straightforward constraints on the parameterization of the canonical models given directly from the definition of discrepancy set:

$$f_X^1(r_X) = f_X^*(r_X), P^1(r_X) = P^*(r_X), \quad X \notin \boldsymbol{\Delta}_{1*}$$
$$f_Z^1(y, r_Z) = f_Z^*(y, r_Z), P^1(r_Z) = P^*(r_Z), \quad Z \notin \boldsymbol{\Delta}_{1*}$$

for any input $x, y, r_Y, r_X, r_Z$. □

The next lemma formalizes the observation made in the example above, showing that if a pair SCMs and a pair of associated canonical models induce the same distributions and causal diagram, their discrepancies must also agree.

**Lemma 2.** *For a pair of SCMs $M^i, M^j$ ($i, j \in \{*, 1, 2, \ldots, T\}$) defined over $\boldsymbol{V}$ with discrepancy set $\Delta_{ij} \subseteq \boldsymbol{V}$, let $\mathcal{N}^i, \mathcal{N}^j$ be associated canonical SCMs that induce the same causal graphs and entail the same distributions over $\boldsymbol{V}$. Then the discrepancy sets of the pairs of SCMs and canonical SCMs must agree, i.e. $V \in \Delta_{ij}$ if and only if either $f_V^{N^i} \neq f_V^{N^j}$, or $P^{N^i}(u_V) \neq P^{N^j}(u_V)$.*

*Proof.* Let $V \in \Delta_{ij}$, and fix $M^i, M^j$ such that $P^{M^i}(v \mid do(pa_V)) \neq P^{M^j}(v \mid do(pa_V))$. This is possible since the interventional probabilities are parameterized by the mechanism of $V$ which could vary across $M^i, M^j$. Assume for a contradiction that $f_V^{N^i} = f_V^{N^j}$ and $P^{N^i}(u_V) = P^{N^j}(u_V)$ for two canonical models $N^i, N^j$ constructed to match all $L_3$ statements induced by $M^i, M^j$. This implies in particular that $P^{\mathcal{N}^i}(v \mid do(pa_V)) = P^{\mathcal{N}^j}(v \mid do(pa_V))$ and therefore $\mathcal{N}^i, \mathcal{N}^j$ do not induce the same probabilities as $M^i, M^j$. This contradicts the assumption that the pair of canonical SCMs matches the pair of SCMs in all $L_3$ statements.

For the converse, we proceed similarly. For fixed $M^i, M^j$, assume for a contradiction that $f_V^{N^i} \neq f_V^{N^j}$, or $P^{N^i}(u_V) \neq P^{N^j}(u_V)$ such that $P^{\mathcal{N}^i}(v \mid do(pa_V)) \neq P^{\mathcal{N}^j}(v \mid do(pa_V))$ for two canonical models $N^i, N^j$ constructed to match all $L_3$ statements induced by $M^i, M^j$, but nevertheless $V \notin \Delta_{ij}$. The discrepancy set ensures that $P^{M^i}(v \mid do(pa_V)) = P^{M^i}(v \mid do(pa_V))$ but the same relation is not true for $N^i, N^j$ as $P^{\mathcal{N}^i}(v \mid do(pa_V)) \neq P^{\mathcal{N}^j}(v \mid do(pa_V))$ by assumption and therefore $\mathcal{N}^i, \mathcal{N}^j$ do not induce the same probabilities as $M^i, M^j$. This contradicts the assumption that the pair of canonical SCMs matches the pair of SCMs in all $L_3$ statements. $\square$

**Lemma 3.** *Consider a system of multiple SCMs $\mathbb{M} : \{\mathcal{M}^1, \mathcal{M}^2, \ldots, \mathcal{M}^K, \mathcal{M}^*\}$ that induces a selection diagram and entails the source distributions $\mathbb{P} : \{P^1, P^2, \ldots, P^K, P^*\}$ over the variables $\boldsymbol{V}$. Then there exists a system of canonical SCM $\mathbb{N} : \{\mathcal{N}^1, \mathcal{N}^2, \ldots, \mathcal{N}^K, \mathcal{N}^*\}$ such that*

1. *$\mathbb{M}$ and $\mathbb{N}$ are associated with the same set of causal diagrams and selection diagrams.*

2. *For any set of counterfactual variables $\boldsymbol{Y_x}, \ldots, \boldsymbol{Z_w}$, $P^{M^*}(\boldsymbol{Y_x}, \ldots, \boldsymbol{Z_w}) = P^{\mathcal{N}^*}(\boldsymbol{Y_x}, \ldots, \boldsymbol{Z_w})$.*

*Proof.* For (1), Thm. 2.4 [42] gives that SCMs $\mathbb{M} : \{\mathcal{M}^1, \mathcal{M}^2, \ldots, \mathcal{M}^K, \mathcal{M}^*\}$ and canonical SCMs $\mathbb{N} : \{\mathcal{N}^1, \mathcal{N}^2, \ldots, \mathcal{N}^K, \mathcal{N}^*\}$ induce the same causal diagrams. Lem. 2 gives that for every pair of SCMs $M^i, M^j$ ($i, j \in \{*, 1, 2, \ldots, T\}$), their discrepancy set is the same as that of $\mathcal{N}^i, \mathcal{N}^j$ ($i, j \in \{*, 1, 2, \ldots, T\}$). As selection diagrams are constructed deterministically from causal diagrams and discrepancy sets, $\mathbb{M}$ and $\mathbb{N}$ must share the same set of selection diagrams.

(2) is given by Thm. 2.4 [42]. $\square$

**Theorem 1 (restated).** *Consider a system of multiple SCMs $\mathcal{M}^1, \mathcal{M}^2, \ldots, \mathcal{M}^K, \mathcal{M}^*$ that induces the selection diagram $\mathcal{G}^{\boldsymbol{\Delta}}$ and entails the source distributions $P^1, P^2, \ldots, P^K$ and the target distribution $P^*$ over the variables $\boldsymbol{V}$. Let $\psi(P^*) \in [0, 1]$ be the target quantity. Consider the following optimization scheme:*

$$\hat{q}_{\max} = \max_{\mathcal{N}^1, \mathcal{N}^2, \ldots, \mathcal{N}^*} \psi(P^{\mathcal{N}^*}) \tag{48}$$

$$s.t. \ P^{\mathcal{N}^i}(r_V) = P^{\mathcal{N}^j}(r_V), \qquad \forall i, j \in \{1, 2, \ldots, K, *\} \quad \forall V \notin \Delta_{i,j}$$

$$P^{\mathcal{N}^i}(\boldsymbol{v}) = P^i(\boldsymbol{v}) \qquad \forall i \in \{1, 2, \ldots, K, *\},$$

*where each $\mathcal{N}^i$ is a canonical model characterized by a joint distribution over $\{R_V\}_{V \in \boldsymbol{V}}$. The value of the above optimization, namely $\hat{q}_{\max}$, is a tight upper-bound for the quantity $\psi(P^*)$ among all tuples of SCMs that induce the selection diagram and entail the source distributions at hand.* $\square$

*Proof.* Note that,

$$\hat{q}_{\max} = \max_{\mathcal{M}^1, \mathcal{M}^2, \ldots, \mathcal{M}^*} \psi(P^{\mathcal{M}^*}) \tag{49}$$

$$\text{s.t. } P^{\mathcal{M}^i}(\boldsymbol{u}_V) = P^{\mathcal{M}^j}(\boldsymbol{u}_V), f_V^{\mathcal{M}^i} = f_V^{\mathcal{M}^j}, \quad \forall i,j \in \{1, 2, \ldots, K, *\} \quad \forall V \notin \Delta_{i,j}$$

$$P^{\mathcal{M}^i}(\boldsymbol{v}) = P^i(\boldsymbol{v}) \qquad \forall i \in \{1, 2, \ldots, K, *\},$$

is a tight upper bound to the target $\psi(P^{\mathcal{M}^*})$ among all tuples of SCMs that induce the selection diagram and entail the source distributions at hand, by construction. It follows from Lem. 3 that for any tuple of SCMs $\{\mathcal{M}^1, \mathcal{M}^2, \ldots, \mathcal{M}^K, \mathcal{M}^*\}$, that induce the selection diagram and entail the source distributions, there exists a tuple of canonical SCMs $\mathcal{N}^1, \mathcal{N}^2, \ldots, \mathcal{N}^*$, that induce the selection diagram and entail the source distributions such that,

$$P^{M^*}(\boldsymbol{Y_x}, \ldots, \boldsymbol{Z_w}) = P^{\mathcal{N}^*}(\boldsymbol{Y_x}, \ldots, \boldsymbol{Z_w}).$$

The reverse direction of the above equations also holds since a a family of canonical SCMs is an instance of a family of SCMs. This means that solutions for optimization problems in Eq. (48) and Eq. (49) must coincide. $\qquad\square$

## E.1 Proof of Theorem 2

To prove this result, we need to show the following:

1. **Necessity.** Every tuple of NCMs $\Theta$ that are constraint by conditions in Eq. 8 represents a tuple of SCMs that entails $\mathbb{P}$ and induces $\mathcal{G}^{\boldsymbol{\Delta}}$.

2. **Sufficiency.** For every tuple of SCMs $\mathbb{M}$ that entails $\mathbb{P}$ and induces $\mathcal{G}^{\boldsymbol{\Delta}}$, there exists a tuple of NCMs $\Theta$ that admits the constraints in Eq. 8, and for every $i \in \{*, 1, 2, \ldots, K\}$, we have $P(\boldsymbol{y_x}, \boldsymbol{z_w}; \theta^i) = P^{\mathcal{M}^i}(\boldsymbol{y_x}, \boldsymbol{z_w})$, where $\boldsymbol{y_x}, \boldsymbol{z_w}$.

**Necessity.** Consider a tuple of NCMs $\Theta$ that are constraint by the conditions in Eq. 8.

- $\mathcal{G}^{\boldsymbol{\Delta}}$**-consistency.** Since these NCMs are constructed based on the common causal diagram $\mathcal{G}$, they all induce $\mathcal{G}$ (Theorem 2 by Xia et al. [41]). Moreover, the parameter sharing constraint states that $V \notin \Delta_{ij}$ if and only if $\theta_V^i = \theta_V^j$. This implies that the NCMs parameterized by $\Theta$ induce the same domain discrepancy sets as $\mathcal{G}^{\boldsymbol{\Delta}}$. Thus, the selection diagram induced by the NCMs parameterized by $\Theta$ is exactly $\mathcal{G}^{\boldsymbol{\Delta}}$.

- $\mathbb{P}$**-expressivity.** The data likelihood condition for source distribution $P^i(\boldsymbol{v})$ states the following:

$$\theta^i \in \underset{\mathcal{G}-\text{constrained } \theta}{\arg\max} \sum_{\boldsymbol{v} \in D^i} \log P(\boldsymbol{v}; \theta). \tag{50}$$

  For large enough samples size $|D^i| \sim P^i(\boldsymbol{v})$, and enough model complexity in $\theta$, Theorem 1 by Xia et al. [41] shows that there exists a $\mathcal{G}$-constrained NCM $\theta$ that induces the distribution entailed by the true SCM $\mathcal{M}^i$. Thus, by imposing Eq. (50) we assure that $P(\boldsymbol{v}; \theta^i) = P^i(\boldsymbol{v})$. By imposing all data likelihood conditions, in the limit of sample size and model complexity, we ensure that the source NCMs induce the source distributions.

In conclusion, the tuple of NCMs are necessarily representing a plausible target domain since (1) they induce $\mathcal{G}^{\boldsymbol{\Delta}}$ and (2) they entail $\mathbb{P}$.

**Sufficiency.** Consider a tuple of SCMs $\mathbb{M} = \langle \mathcal{M}^1, \mathcal{M}^2, \ldots, \mathcal{M}^K, \mathcal{M}^* \rangle$ that induce $\mathcal{G}^{\boldsymbol{\Delta}}$ and entail $\mathbb{P}$. Theorem 1 by Xia et al. [41] shows that for every SCM $\mathcal{M}$ that induces $\mathcal{G}$, there exists a $\mathcal{G}$-constraint NCM parameterized by $\theta$ such that $P^{\mathcal{M}}(\boldsymbol{v}) = P(\boldsymbol{v}; \theta)$ (as a consequence of L3-consistency). The proof is constructive, and for every $V \in \boldsymbol{V}$ the construction of the neural network $\theta_V$ depends on (1) the function $f_V$ and (2) the distribution $P^{\mathcal{M}}(\boldsymbol{u}_V)$.

Consider two SCMs $\mathcal{M}^i, \mathcal{M}^j$ ($i, j \in \{*, 1, 2, \ldots, K\}$) that induce domain discrepancy set $\Delta_{ij}$. Follow the construction by Xia et al. [41] to obtain the corresponding NCMs parameterized by $\theta^i, \theta^j$. For every $V \notin \Delta_{i,j}$, we have, $\theta_V^i = \theta_V^j$ since the construction depends on $f_V^i = f_V^j$ and $P^{\mathcal{M}^i}(\boldsymbol{u}_V) = P^{\mathcal{M}^j}(\boldsymbol{u}_V)$. Thus, the domain discrepancy set induced by $\theta^i, \theta^j$ matches with $\Delta_{i,j}$ induced by the

SCMs $\mathcal{M}^i, \mathcal{M}^j$. Therefore, By constructing the NCM $\theta^i$ from $\mathcal{M}^i$ ($i \in \{*, 1, 2, \ldots, K\}$), we are guaranteed to have a tuple of NCMs $\mathbb{N}$ that (1) induce $\mathcal{G}^{\Delta}$ and (2) entails $\mathbb{P}$.

**Partial-TR via NCMs.** Due to necessity and sufficiency above, we conclude that a tuples of NCMs satisfies the parameter sharing and data likelihood conditions stated in Eq. 8, if and only if there exists a tuple of SCMs $\mathbb{M}$ that induce $\mathcal{G}^{\Delta}$ and entail $\mathbb{P}$ such that $P(\boldsymbol{v}; \theta^i) = P^{\mathcal{M}^i}(\boldsymbol{v})$ for all $i \in \{*, 1, 2, \ldots, K\}$. Therefore, by solving the following optimization problem,

$$\hat{\Theta} \in \underset{\Theta:\langle \theta^1, \theta^2, \ldots, \theta^K, \theta* \rangle}{\arg\max} \sum_{\boldsymbol{w}} \psi(\boldsymbol{w}) \cdot \sum_{\boldsymbol{v} \backslash \boldsymbol{w}} P(\boldsymbol{v}; \theta^*) \tag{51}$$

$$\text{s.t. } \theta_V^i = \theta_V^j, \qquad \forall i, j \in \{1, 2, \ldots, K, *\} \quad \forall V \notin \Delta_{i,j}$$

$$\theta^i \in \underset{\theta}{\arg\max} \sum_{\boldsymbol{v} \in D^i} \log P(\boldsymbol{v}; \theta), \quad \forall i \in \{1, 2, \ldots, K\}.$$

we achieve a tight upper-bound for the query $\mathbb{E}_{P*}[\psi(\boldsymbol{W})]$ w.r.t. $\mathcal{G}^{\Delta}, \mathbb{P}$. □

### E.2 Proof of Proposition 1

Consider the objective of Theorem 2;

$$\hat{\Theta} \in \underset{\Theta:\langle \theta^1, \theta^2, \ldots, \theta^K, \theta* \rangle}{\arg\max} \sum_{\boldsymbol{w}} \psi(\boldsymbol{w}) \cdot \sum_{\boldsymbol{v} \backslash \boldsymbol{w}} P(\boldsymbol{v}; \theta^*) \tag{52}$$

$$\text{s.t. } \theta_V^i = \theta_V^j, \qquad \forall i, j \in \{1, 2, \ldots, K, *\} \quad \forall V \notin \Delta_{i,j}$$

$$\theta^i \in \underset{\theta}{\arg\max} \sum_{\boldsymbol{v} \in D^i} \log P(\boldsymbol{v}; \theta), \quad \forall i \in \{1, 2, \ldots, K\}.$$

**No need to parameterize non-ancestors of $W$.** Let $\boldsymbol{T} = \boldsymbol{V} \backslash An_{\mathcal{G}*}(\boldsymbol{W})$. By applying Rule 3 of $\sigma$-calculus [9] we realize that,

$$P(\boldsymbol{w}; \theta_{\boldsymbol{V} \backslash \boldsymbol{T}}^*, \theta_{\boldsymbol{T}}^*) = P(\boldsymbol{w}; \theta_{\boldsymbol{V} \backslash \boldsymbol{T}}^*, \tilde{\theta}_{\boldsymbol{T}}). \tag{53}$$

The latter indicates that the parameters $\{\theta_T^*\}_{T \in \boldsymbol{T}}$ are irrelevant to the joint distribution $P(\boldsymbol{w})$, and therefore, can be dropped from the NCMs used for partial transportability of $\mathbb{E}_{P*}[\psi(\boldsymbol{W})] = \sum_{\boldsymbol{w}} P^*(\boldsymbol{w}) \cdot \psi(\boldsymbol{w})$.

Let $\boldsymbol{A} = An_{G*}(\boldsymbol{W})$. We drop the non-ancestors, and rewrite the objective as follows:

$$\hat{\Theta_{\boldsymbol{A}}} \in \underset{\Theta_{\boldsymbol{A}}:\langle \theta_{\boldsymbol{A}}^1, \theta_{\boldsymbol{A}}^2, \ldots, \theta_{\boldsymbol{A}}^K, \theta_{\boldsymbol{A}}^* \rangle}{\arg\max} \sum_{\boldsymbol{w}} \psi(\boldsymbol{w}) \cdot \sum_{\boldsymbol{a} \backslash \boldsymbol{w}} P(\boldsymbol{a}; \theta^*) \tag{54}$$

$$\text{s.t. } \theta_V^i = \theta_V^j, \qquad \forall i, j \in \{1, 2, \ldots, K, *\} \quad \forall V \notin \Delta_{i,j}$$

$$\theta_{\boldsymbol{A}}^i \in \underset{\theta}{\arg\max} \sum_{\boldsymbol{a} \in D^i} \log P(\boldsymbol{a}; \theta_{\boldsymbol{A}}), \quad \forall i \in \{1, 2, \ldots, K\}.$$

Next, we add the likelihood terms to the main objective regularized by a coefficient $\Lambda$ to achieve a single-objective optimization.

$$\hat{\Theta_{\boldsymbol{A}}} \in \underset{\Theta_{\boldsymbol{A}}:\langle \theta_{\boldsymbol{A}}^1, \theta^2, \ldots, \theta^K, \theta* \rangle}{\arg\max} \sum_{\boldsymbol{w}} \psi(\boldsymbol{w}) \cdot \sum_{\boldsymbol{a} \backslash \boldsymbol{w}} P(\boldsymbol{a}; \theta^*) + \Lambda \cdot \sum_{i=1}^{K} \sum_{\boldsymbol{a} \in D^i} \log P(\boldsymbol{a}; \theta_{\boldsymbol{A}}^i) \tag{55}$$

$$\text{s.t. } \theta_V^i = \theta_V^j, \qquad \forall i, j \in \{1, 2, \ldots, K, *\} \quad \forall V \notin \Delta_{i,j}$$

For $\Lambda \to \infty$, the new optimization problem matches with that of Thm. 2. Now, we focus on the likelihood expression, and rewrite it following a causal order of $\mathcal{G}^*$, namely, $A_1 \prec A_2 \prec \cdots \prec A_N$.

$$\log P(\boldsymbol{a}; \theta_{\boldsymbol{A}}^i) = \sum_{l=1}^{N} \log P(a_l \mid a_{l-1}, \ldots, a_1; \theta_{\boldsymbol{A}}^i) \qquad \text{(factorization)} \qquad (56)$$

$$= \sum_{l=1}^{N} \log \mathbb{E}_{\boldsymbol{U_A}}[P(a_l \mid v_{l-1}, \ldots, v_1, \boldsymbol{U}; \theta_{\boldsymbol{A}}^i)] \qquad \text{(conditioning on } \boldsymbol{U}) \qquad (57)$$

$$= \sum_{l=1}^{N} \log \mathbb{E}_{\boldsymbol{U}_{A_l}}[P(a_l \mid pa_{A_l}, \boldsymbol{U}_{A_l}; \theta^i)] \qquad \text{(Rule 1 of do-calc)} \qquad (58)$$

$$= \sum_{l=1}^{N} \log \mathbb{E}_{\boldsymbol{U}_{A_l}}[P(a_l \mid pa_{A_l}, \boldsymbol{U}_{A_l}; \theta_A^i)] \qquad \text{(Rule 3 of do-calc)} \qquad (59)$$

Let $\{\boldsymbol{A}_j\}_{j=1}^m$ be the c-components of $\mathcal{G}_{[\boldsymbol{A}]}^*$, which is the graph induced by nodes $\boldsymbol{A}$. We rewrite the above objective in terms of the c-factors:

$$\log P(\boldsymbol{a}; \theta_{\boldsymbol{A}}^i) = \sum_{j=1}^{m} \sum_{A \in \boldsymbol{A}_j} \log \mathbb{E}_{\boldsymbol{U_A}}[P(a \mid pa_A, \boldsymbol{U}_A; \theta_A^i)] \qquad \text{(c-factor decomp.)} \qquad (60)$$

$$= \sum_{j=1}^{m} \log \prod_{A \in \boldsymbol{A}_j} \mathbb{E}_{\boldsymbol{U_A}}[P(a \mid pa_A, \boldsymbol{U}_A; \theta_A^i)] \qquad \text{(sum-of-log to log-of-prod)} \qquad (61)$$

$$= \sum_{j=1}^{m} \log \mathbb{E}_{\boldsymbol{U}_{\boldsymbol{A}_j}}[\prod_{A \in \boldsymbol{A}_j} P(a \mid pa_A, \boldsymbol{U}_A; \theta_A^i)] \qquad \text{(mutually indep. } \boldsymbol{U}_A) \qquad (62)$$

$$= \sum_{j=1}^{m} \log P(\boldsymbol{a}_j \mid do(pa_{\boldsymbol{A}_j}); \theta_{\boldsymbol{A}_j}^i) \qquad \text{(trunc. fact. prod.)} \qquad (63)$$

From the last expression, we can observe that the NCM parameterization is modular w.r.t. the c-components, as Rahman et al. [28] also discusses. We rewrite the full optimization program again:

$$\hat{\boldsymbol{\Theta}_{\boldsymbol{A}}} \in \underset{\boldsymbol{\Theta_A}:\langle\theta_{\boldsymbol{A}}^1, \theta^2, \ldots, \theta^K, \theta^*\rangle}{\arg\max} \sum_{\boldsymbol{a}} \exp\{\sum_{j=1}^{m} \log P(\boldsymbol{a}_j \mid do(pa_{\boldsymbol{A}_j}); \theta_{\boldsymbol{A}_j}^*)\} \cdot \psi(\boldsymbol{a}) \qquad (64)$$

$$+ \Lambda \cdot \sum_{i=1}^{K} \sum_{j=1}^{m} \sum_{\boldsymbol{a}_j \in D^i} \log P(\boldsymbol{a}_j \mid do(pa_{\boldsymbol{A}_j}); \theta_{\boldsymbol{A}_j}^i) \qquad (65)$$

$$\text{s.t. } \theta_V^i = \theta_V^j, \qquad \forall i, j \in \{1, 2, \ldots, K, *\} \quad \forall V \notin \Delta_{i,j}$$

Let $\boldsymbol{A}_j$ be a c-component that $S_i$ is not pointing to it in $\mathcal{G}^{\boldsymbol{\Delta}}$, i.e., $\boldsymbol{A}_j \cap \Delta_i = \varnothing$. The latter means that the parameter sharing $\theta_V^* = \theta_V^i$ is enforced for all $V \in \boldsymbol{A}_j$; we call these parameters $\theta_{\boldsymbol{A}_j}^{i,*}$. We notice that $\theta_{\boldsymbol{A}_j}^{i,*}$ only appears through the term $\log P(\boldsymbol{a}_j \mid do(pa_{\boldsymbol{A}_j}); \theta_{\boldsymbol{A}_j})$ in the score function; once in the main objective as $\theta_{\boldsymbol{A}_j}^*$ and once in the regularizer as $\theta_{\boldsymbol{A}_j}^i$. For $\Lambda \to \infty$, the regularizer enforces $\theta_{\boldsymbol{A}_j}^{i,*}$ to satisfy the following criterion:

$$\theta_{\boldsymbol{A}_j}^{i,*} \in \underset{\theta_{\boldsymbol{A}_j}}{\arg\max} \sum_{\boldsymbol{a}_j \in D^i} \log P(\boldsymbol{a}_j \mid do(pa_{\boldsymbol{A}_j}); \theta_{\boldsymbol{A}_j}) \qquad (66)$$

This criterion is in fact an interventional (L2) constraint [41] enforced on $\theta_{\boldsymbol{A}_j}^{i,*}$ that requires $\theta_{\boldsymbol{A}_j}^{i,*}$ to approximate $P^i(\boldsymbol{a}_j \mid do(pa_{\boldsymbol{A}_j}))$ using the observational data $D^i$. Since $P^i(\boldsymbol{a}_j \mid do(pa_{\boldsymbol{A}_j}))$ is a complete c-factor, it is identifiable from $P^i(\boldsymbol{a}_j, pa_{\boldsymbol{A}_j})$ [36]. Therefore, by increasing the sample size $|D^i| \to \infty$ and the model complexity of $\theta_{\boldsymbol{A}_j}^{i,*}$, satisfying the criterion in Eq. 66 guarantees arbitrarily accurate approximation of the interventional quantities $P^i(\boldsymbol{a}_j \mid do(pa_{\boldsymbol{A}_j}))$ [41]. This implies that we can replace the terms involving the parameters $\theta_{\boldsymbol{A}_j}^{i,*}$ with any consistent approximation

of $P^i(\boldsymbol{a}_j \mid do(pa_{\boldsymbol{A}_j}))$ as constants. To get the approximation, we are free to use any probabilistic model and architecture depending on the context; this includes the option to train the NCM parameters $\theta_{\boldsymbol{A}_j}^{i,*}$ in the pre-training.

This adjustment gets us to the exact procedure pursued in Algorithm 1, thus proves consistency of it with what we would achieve via Theorem 2. $\qquad \square$

### E.3 Proof of Theorem 3

For this proof, it is useful to define the worst-case risk w.r.t. the selection diagram and the source distributions.

**Definition 8** (Worst-case risk). *For selection diagram $\mathcal{G}^{\boldsymbol{\Delta}}$ and source distributions $\mathbb{P}$, the worst-case risk of classifer $h : \Omega_{\boldsymbol{X}} \to \Omega_Y$ is denoted by $R_{\mathcal{G}^{\boldsymbol{\Delta}},\mathbb{P}}(h)$ and defined as the solution of partial transportation task for the query $\mathbb{E}_{P*}[\mathcal{L}(Y, h(\boldsymbol{X}))]$, where $\mathcal{L}(y, \hat{y})$ is a loss function. Formally,*

$$R_{\mathcal{G}^{\boldsymbol{\Delta}},\mathbb{P}}(h) := \max_{\text{tuple of SCMs } \mathbb{M}_0 \text{ that entails } \mathbb{P} \text{ \& induces } \mathcal{G}^{\boldsymbol{\Delta}}} R_{P^{\mathcal{M}_0^*}}(h). \qquad (67)$$

$\qquad \square$

**Theorem 3** (restated). *For discrete $\boldsymbol{X}, Y$ CRO terminates. Furthermore, for large enough data across all source domain, the worst-case risk of CRO is at most $\epsilon$ away from the worst-case optimal classifier w.r.t. selection diagram $\mathcal{G}^{\boldsymbol{\Delta}}$ and source data $\mathbb{P}$. Formally,*

$$\lim_{n \to \infty} P(R_{\mathcal{G}^{\boldsymbol{\Delta}},\mathbb{P}}(h_n^{\text{CRO}}) - \min_{h:\Omega_{\boldsymbol{X}} \to \Omega_Y} R_{\mathcal{G}^{\boldsymbol{\Delta}},\mathbb{P}}(h) > \epsilon) \to 0 \qquad (68)$$

*where $h_n^{\text{CRO}} := \text{CRO}(\mathbb{D}_n, \mathcal{G}^{\boldsymbol{\Delta}})$, and $\mathbb{D}_n = \langle D^1, D^2, \ldots, D^K \rangle$ is a collection of datasets that each contain at least $n$ datapoints.* $\qquad \square$

*Proof.* Soundness of CRO relies on consistency of Neural-TR (Alg. 1 as a subroutine; we pick the data size large enough to satisfy this condition according to Theorem 2.

**Termination.** Let $\hat{\theta}_1^*, \hat{\theta}_2^*, \ldots$ be the sequence of target NCMs produced during the runtime of CRO, and let $h_1, h_2, \ldots$ be the sequence of classifiers obtained after each iteration. Let $\Pi$ denote the space of all distributions over $\boldsymbol{X}, Y$. For discrete $\boldsymbol{X}, Y$, the space $\Pi$ is a compact subspace of some Euclidean space. Thus, every sequence in $\Pi$ has a convergent subsequence, especially the sequence $\{P(\boldsymbol{x}, y; \hat{\theta}_m^*)\}_m \subset \Pi$; let $\{P_l\}_l$ be this convergent subsequence. Every convergent subsequence is Cauchy, which means,

$$\forall \tau > 0 \quad \exists n > 0 \quad \forall l, l' > n : d(P_l - P_{l'}) < \tau, \qquad (69)$$

where $d$ is an appropriate metric over the probability space. Choose $\tau$ small enough w.r.t. the convergence tolerance $\delta > 0$ to ensure,

$$\forall P, P' \text{ where } d(P, P') < \tau \implies \forall h : \Omega_{\boldsymbol{X}} \to \Omega_Y \quad |R_P(h) - R_{P'}(h)| \leqslant \delta. \qquad (70)$$

The above is possible, since the mapping $R_P(h)$ is a bounded and continuous mapping on the space $\Pi$. Now, we are guaranteed to find an index $l$ such that,

$$|R_{P(\boldsymbol{x},y;\hat{\theta}_{l+1}^*)}(h_l) - R_{P(\boldsymbol{x},y;\hat{\theta}_l^*)}(h_l)| < \delta. \qquad (71)$$

Notice that by definition, $\hat{\theta}_{l+1}^*$ is obtained by Neural-TR (Alg. 1) to attain the worst-case risk of $h_l$, i.e.,

$$R_{P(\boldsymbol{x},y;\hat{\theta}_{l+1}^*)}(h_l) = R_{\mathcal{G}^{\boldsymbol{\Delta}},\mathbb{P}}(h_l). \qquad (72)$$

Moreover,

$$R_{P(\boldsymbol{x},y;\hat{\theta}_l^*)}(h_l) \leqslant \max_{i \in \{1,2,\ldots,l\}} R_{D_i^*}(h_l). \qquad (73)$$

Putting the last three equations together we have,

$$R_{\mathcal{G}^{\boldsymbol{\Delta}},\mathbb{P}}(h_l) \leqslant \max_{i \in \{1,2,\ldots,l\}} R_{D_i^*}(h_l) + \delta, \qquad (74)$$

which invokes the termination.

**Worst-case optimality.** Suppose $h^{\text{CRO}}$ is returned by CRO, and let $h^*$ be the true worst-case optimal classifier defined as,

$$h^* \in \min_{h:\Omega_{\boldsymbol{X}} \to \Omega_Y} R_{\mathcal{G}^\Delta, \mathbb{P}}(h). \tag{75}$$

Let $\mathbb{D}$ denote the collection of datasets collected by the algorithm before termination. We know that $h^{\text{CRO}}$ is robust to $\mathbb{D}^*$, i.e.,

$$h^{\text{CRO}} \in \arg\min_{h:\Omega_{\boldsymbol{X}} \to \Omega_Y} \max_{D \in D^*} R_D(h) \implies \max_{D \in D^*} R_D(h^{\text{CRO}}) \overset{\text{opt. } h^{\text{CRO}}}{\leqslant} \max_{D \in D^*} R_D(h^*) \tag{76}$$

Moreover, every distribution in $\mathbb{D}^*$ is entailed by an NCM that represents a possible target domain. Therefore, the worst-case risk is at least as large as the worst-case empirical risk on the set of distribution $\mathbb{D}$, i.e.,

$$\max_{D \in D^*} R_D(h^*) \leqslant R_{\mathcal{G}^\Delta, \mathbb{P}}(h^*) \tag{77}$$

Since that algorithm has terminated we have,

$$R_{\mathcal{G}^\Delta, \mathbb{P}}(h^{\text{CRO}}) < \max_{D \in D^*} R_D(h^{\text{CRO}}) + \delta. \tag{78}$$

where $\delta > 0$ is the tolerance for the convergence condition in the algorithm. Putting all inequalities together, we have,

$$R_{\mathcal{G}^\Delta, \mathbb{P}}(h^{\text{CRO}}) - \delta \leqslant \max_{D \in D^*} R_D(h^{\text{CRO}}) \tag{79}$$

$$\leqslant \max_{D \in D^*} R_D(h^*) \tag{80}$$

$$\leqslant R_{\mathcal{G}^\Delta, \mathbb{P}}(h^*), \tag{81}$$

which indicates that the worst-case risk of $h^{\text{CRO}}$ is at most $\delta$ larger than the optimal worst-case risk. $\qquad\square$

# F  Broader Impact and Limitations

Our work investigates the design of algorithms and conditions under which knowledge acquired in one domain (e.g., particular setting, experimental condition, scenario) can be generalized to a different one that may be related, but is unlikely to be the same. As alluded to in this paper, under-identifiability issues and the difficulty of stating realistic assumptions that are conducive to extrapolation guarantees are pervasive throughout the data sciences. Our hope is that our analysis with a more surgical encoding of structural differences between domains that allow the empirical investigator to determine whether (and how) her/his understanding of the underlying system is sufficient to support the generalization of prediction algorithm is an important addition towards safe and reliable AI. This approach is not without limitations, however. We have shown that selection diagrams are sufficient to ensure consistent domain generalization (through bounds instead of point estimates) but arguably restrict the analysis to a narrow class of problems as graphs or super-structures need to be defined. This stands in contrast with representation learning methods that operate on higher-dimensional spaces, e.g. text, images, which are difficult to reason about in a causal framework. The trade-off is that guarantees for consistent extrapolation are difficult to define and that one-size-fits-all assumptions are difficult to justify in practice. Partial transportability may be understood as a complementary view-point on this problem, applicable in a different class of problems in which structural knowledge is available implying that non-trivial guarantees for extrapolation can be established. Pushing the boundaries of methods based on causal graphs to reach compelling real-world applications is arguably one the most important frontiers for the causal community as a whole. In this work, there is scope for improving posterior estimation and for introducing assumptions on the class of SCMs that are modelled, e.g. linear Gaussian models, etc., that could lead to efficient predictors in higher-dimensional spaces. Similarly, relaxations of selection diagrams, e.g. in the form of equivalence classes or partially-known graphs, could be developed for applications in domains where knowledge of graph structure is unrealistic.

