# OpenReview forum: "Partial Transportability for Domain Generalization"
_NeurIPS.cc/2024/Conference — NeurIPS 2024 poster_

### Official Review · Reviewer_SwAx · 2024-07-09

**Soundness:** 3
**Presentation:** 3
**Contribution:** 3
**Rating:** 6
**Confidence:** 3

**Summary:**

This paper tackles to problem of transportability in domain generalization. Multiple datasets are provided together with graphical information stipulating which causal mechanisms are shared across domains and which mechanisms are going to remain invariant in the target domain. The goal is then to estimate the error committed by a given classifier in the target domain. More specifically, the authors propose a way to estimate the worst-case error of the classifier given the observed distributions as well as the graphical information. A first result shows how the problem can be simplified using canonical SCMs and a second result shows how Neural Causal Models (NCM) can be used to estimate this bound. Furthermore, an approach named Causal Robust Optimization (CRO) is proposed to learn a classifier that minimizes the worst-case risk. Experiments experiments with the NCM approach to estimate the upper-bound of various classifiers, but does not tackle the CRO approach to learn a robust optimizer.

**Strengths:**

- The paper reads nicely (even for a fairly technical paper) and gives a good background on SCMs and graph terminology for the non-expert. The choice of notation is good and remains consistent throughout the paper.
- The problem appears to be very important, and the approach to tackle it seems principled.
- The approach appears to be novel and interesting. However, I don't have a good knowledge of the literature on transportability and domain generalization and thus my judgment's credibility is limited.
- Many examples are provided to make the discussion more concrete, I enjoyed that.

**Weaknesses:**

- Some theoretical results, in their current form, seems unlikely to be true. Specifically in Theorem 2, I was surprised to see that this result does not mention anything about the expressive power of the NCM, what if the neural networks do not have enough capacity to express the causal mechanisms? Clearly what you would end up with wouldn’t be an upper bound right? I was also surprised to see that the optimization problem is formulated in a finite-sample fashion, but the conclusion concerns the actual expectation (i.e. not finite-sample), this sounds unlikely, no? Don’t you need to take a limit as the dataset grows for this bound to hold?

- Confusion surrounding canonical SCM: At line 181, it is mentioned that [41] showed that every SCM M can be written in a canonical form (with discrete exogenous variables)... However, after quickly skimming through the paper, I could not find something resembling that statement. What am I missing? Overall I thought the discussion on canonical SCM was confusing. I’m left unsure as to which type of SCM can be cast to a canonical form (definition 5) and which paper showed this fact.

- Some section were hard to parse. Mainly lines 270 to 289, including Algorithm 1. I'm not sure I got the point. I was following up to that point. I give some minor suggestions to improve clarity below.

- Algorithm 2 (Causal Robust Optimization) is interesting, but it looks quite costly. Do the authors believe it could be made applicable? I was a bit disappointed to see it was not implemented in the experimental section. It would be nice to have experiments for CRO. Could this approach be applied to the Coloured MNIST dataset?

- The background is very well written, but the end of the paper felt a bit less polished. Since these are the most important part, I believe they deserve more space. For instance, I had a hard time understanding the experiments with colored MNIST.

Minor:
- Example 1: Why using arrows for assignment? The symbol “:=” was used earlier…
- Definition 3: Should S_{ij} be in fact S_i here? (line 125)
- Example 2: Long sequences of “if”s would benefit from some punctuation. The “circled plus sign” is an XOR right?
- Definition 4: Might be helpful to spell out \mathbb{M}_0 completely in (4), since \mathcal{M}^*_0 appears in the expectation but is not defined (the reader has to deduce that this is the target SCM belonging to  \mathbb{M}_0)
- Definition 5: might be useful to end with “where h_V^{(r_V)} is a function from supp_{pa_V} to supp_V”. I’m actually not certain the very last sentence adds something to the definition, since it is already obvious that f_V(pa_V, r_V) is a function of pa_V for all r_V… I’m also a bit confused by this sentence: “The set of endogenous variables V is discrete.” What does it mean for a set to be discrete? Do you mean the endogenous variables are discrete? Also, I’m a bit confused by the way the word “support” is used here. Usually it means the set of values for which the random variable has positive probability (at least for discrete variables that’s the definition), but here I believe this is not how “support” is used, correct?
- Theorem 1: I believe the first constraint should not include the case i = *, no? I thought P* was not observed. In fact, you don’t have the case i=* on line 248 and in Theorem 2
- Figure 1 is unclear, what does the color mean?
- Line 353: Bayesian inference procedure?

**Questions:**

See above.

**Limitations:**

I didn't find any discussion of the limitations of the approach in the paper.

---

> ### Author Rebuttal · Authors · 2024-08-06
>
> We appreciate the time and effort for reviewing our paper and the positive assessment that our paper “reads nicely”, “the problem appears important” and that our approach is “principled”. Please find below our response to the review, and let us know if you have any further concerns.
>
> > **Q 1.** *”Theorem 2, expressive power of the NCM. Don’t you need to take a limit as the dataset grows for this bound to hold?”*
>
> Thank you for raising this point. The claim of Theorem 2 holds in the limit of data from the source domains, and increasing expressivity of the feed-forward Neural Networks that parameterize the NCMs. This fact was more clearly stated in the proof of Theorem 2 (line 610 and 615) and we will correct the statement in the main body of the paper, thank you for pointing this out.
>
> As an additional observation, we could note that with finite data (provided that the Neural Networks are expressive enough) we would achieve a valid upper bound, since the set of SCMs compatible with finite data is a superset of the set of SCMs compatible with the underlying distributions. The important distinction, however, is that with finite data the bound will not be tight in general.
>
> > **Q. 2.** *”Confusion surrounding canonical SCM: At line 181, it is mentioned that [41] showed that every SCM M can be written in a canonical form (with discrete exogenous variables)... However, after quickly skimming through the paper, I could not find something resembling that statement. What am I missing? Overall I thought the discussion on canonical SCM was confusing. I’m left unsure as to which type of SCM can be cast to a canonical form (definition 5) and which paper showed this fact.”*
>
> Thank you for your careful reading. The reference [41] should be labeled as [42] referring to Zhang et al. “Partial counterfactual identification from observational and experimental data”. Specifically, Theorem 2.4 shows that for any SCM defined over a set of discretely-valued (and finite) endogenous variables, there exists a canonical SCM (as defined in our Def. 5) that is equally expressive, i.e., it generates the same observational, interventional, and counterfactual distributions.
>
> To answer your question more directly: any SCM over discretely-valued (and finite) endogenous variables can be cast as a canonical model without loss of generality. Having said that, and since this is a fundamental part of the background, we will add a new appendix introducing these results to make the presentation more self-contained.
>
> > **Q 3.** *”Some section were hard to parse. Mainly lines 270 to 289, including Algorithm 1. I'm not sure I got the point. I was following up to that point. I give some minor suggestions to improve clarity below.”*
>
> Lines 270 to 289 refer to Example 4 that we use to convey the fact that joint distributions $P^*(\boldsymbol V)$ may be factorized into several terms (as in line 217). Given the assumptions encoded by the selection diagram and the source data, some of those terms may be point identifiable and evaluated uniquely through the source data, i.e. no bounding necessary, while others may not be evaluated uniquely. More broadly, the point of example 4 is to show that the parameter space can be cleverly decoupled and the computational cost of the optimization problem can be significantly improved: since only a subset of the conditional distributions need to be parameterized and optimized. This component of our work is a contribution over the existing work by Xia et al. 2021 ([35] in the paper) where the entire SCMs are parameterized by neural networks rather than only non-identifiable components.
>
> This observation motivates Alg. 1, that (1) decomposes the query, (2) computes the identifiable components, and (3) parameterizes the components that are not point identifiable, and then proceeds with NCM optimization. We have added a more thorough discussion and illustration of Alg. 1 in the Appendix.
>
> > **Q 4.** *”Algorithm 2 (Causal Robust Optimization) is interesting, but it looks quite costly. Do the authors believe it could be made applicable? I was a bit disappointed to see it was not implemented in the experimental section. It would be nice to have experiments for CRO. Could this approach be applied to the Coloured MNIST dataset?”*
>
> We appreciate the suggestion and we have now implemented CRO and conducted experiments on synthetic examples (Example 2 and 3) and Colored MNIST. Please find all performance details in the global rebuttal and attached pdf. We have observed CRO to terminate in less than 4 iterations in all experiments (3 iterations for Colored MNIST) which we believe makes it very practical even in higher-dimensional problems such as image classification.
>
> Indeed, we did not provide any guarantees for the number of iterations it takes for CRO to terminate; the concern about efficiency is quite valid from the theoretical perspective and a deeper analysis would be interesting.
>
> > **Q 5.** *”The background is very well written, but the end of the paper felt a bit less polished. Since these are the most important part, I believe they deserve more space. For instance, I had a hard time understanding the experiments with colored MNIST.”*
>
> We appreciate the comment. With the additional experiments on CRO and additional space in the camera-ready version of the paper, we will better describe the setting and results of our experiments. Please find a recap of CRO and the colored-MNIST example in the global rebuttal.
>
> > **Q 6.** *”Minor comments.”*
>
> Thank you for pointing these out. We will make sure to clarify / correct as required.
>
> > **Q 7.** *"I didn't find any discussion of the limitations of the approach in the paper."*
>
> Broader impact and limitations are discussed in Appendix B.

---

> > ### Author Response · Authors · 2024-08-13
> >
> > With the discussion period coming to its end, we were wondering whether you had a chance to check our rebuttal. We hope to have answered all concerns to your satisfaction. If not, please don't hesitate to get in touch if there is any concern we could still help to clarify.
> >
> > Thank you again for your time and attention.

---

> > > ### Comment · Reviewer_SwAx · 2024-08-13
> > >
> > > Thanks for the very detailed rebuttal. This is good work so I believe this work should be accepted. I maintain my score.

---

### Official Review · Reviewer_fw9Z · 2024-07-13

**Soundness:** 4
**Presentation:** 3
**Contribution:** 2
**Rating:** 6
**Confidence:** 1

**Summary:**

The paper extends the formulation of canonical models to encode the constraints for the transportability tasks. Then it adapts Neural Causal Models for the transportability task and introduces an iterative method Causal Robust Optimization to find a predictor with the best worst-case risk.

**Strengths:**

- The paper presents a unified framework for addressing transportability problems using canonical models and neural causal models. These transportability problems are broad and highly relevant in the field of machine learning.
- I appreciate the examples illustrated throughout the main text, as they greatly help the reader understand the definitions and intuitions.
- The theorems are solid and the experiments align well with the theories.

**Weaknesses:**

Disclaimer: I am not familiar with this field, so my observations and suggestions are based on my general understanding on the paper.

- Some definitions are not provided, making it difficult for readers to understand certain parts of the paper. For example line 133, 134 $\bigoplus$.

Minor:
1. Page 1 line 9, “such as”.

**Questions:**

1. What does $\bigoplus$ mean?
2. Could you provide some experimental results for CRO and also compare it with other algorithms?

**Limitations:**

The authors have adequately addressed the limitations.

---

> ### Author Rebuttal · Authors · 2024-08-06
>
> We appreciate the positive assessment of our work, thank you. The following response answers each question sequentially. We would be happy to expand on it if needed.
>
> > **Q 1.** *”Some definitions are not provided, making it difficult for readers to understand certain parts of the paper. For example line 133, 134 (xor operator)”*
>
> The $\bigoplus$ denotes the xor operator: $A\bigoplus B$ evaluates to 1 if $A\neq B$ and evaluates to 0 if $A=B$.
>
> > **Q 2.** *”Could you provide some experimental results for CRO and also compare it with other algorithms?”*
>
> Yes, thank you for this suggestion. We have added experimental results for CRO for all experiments, including for the synthetic simulations (Examples 2 and 3) and for Colored MNIST. This analysis is provided in more detail in the global rebuttal and attached pdf, with figures describing the learning process of CRO as well as explicit comparisons with baseline algorithms. As a summary, recall that the CRO algorithm uses Neural-TR as a subroutine to iteratively enhance a classifier and arrive at the best worst-case classifier. In our experiments, CRO converges in 4 iterations or less. In both the simulated and the Colored MNIST experiment we verify in every case that CRO achieves the best possible worst-case performance (known in these examples by construction), as suggested by Thm. 3.
>
> We also note that the poor performance of the baseline algorithms should not be blatantly compared to that of CRO, since CRO has access to background information that can not be communicated with the baseline algorithms. In this sense, CRO can be viewed as a meta-algorithm that operates with a broad range of assumptions encoded in a certain format (i.e., the selection diagram), while the baseline algorithms lack this capacity, and therefore, CRO is able to find the theoretically optimal classifier for domain generalization while the baseline algorithms fail to achieve that and perform poorly in the worst-case scenario.

---

> > ### Comment · Reviewer_fw9Z · 2024-08-07
> > **After Rebuttal**
> >
> > Thanks to the authors for their reply. I appreciate the experiments they add. I will keep my score.

---

### Official Review · Reviewer_S6HF · 2024-07-13

**Soundness:** 3
**Presentation:** 2
**Contribution:** 3
**Rating:** 6
**Confidence:** 2

**Summary:**

This paper studies the problem of domain generalization through the lens of partial transportability, and introduces some new results for bounding the value of a functional of the target distribution, given data from source domains and assumptions about the data generating mechanisms.  Authors adapt existing parameterization schemes such as Neural Causal Models to encode the structural constraints necessary for cross-population inference. Some experiments and examples are also provided.

**Strengths:**

- Introducing partial transportability and canonical SCM to domain generalization is novel.

- The theoretic results are sound and have complete proofs. They are potentially useful.

**Weaknesses:**

- The paper organization and presentation seem complicated to convey the idea of the paper (see below).
- It is not clear how this bound can in turn help improve general DG problems.

**Questions:**

Based on my understanding, authors consider bounding the queries of generalization errors in non-transportable settings, by obtaining a bound for the worst-case losses. Authors use the formulation of canonical  models as a way of encoding constraints to derive such a bound.

1. I believe there are existing works that assume some prior information regarding target domain and then derive bounds for the target loss in DG problems. If my understanding is correct, then I would like to treat the current work as **an alternative way** of encoding information about the target domain. Then the current way of presentation and organization indeed make  it complicated to understand the paper. In particular, I feel that examples 1 and 2 somehow distract reader's attention, while it is better to directly formulate the central problem of the paper.

2. Other questions:
   1. line 125:  what does $j$ indicate in the notation $S_{ij}$?
   2. definition 4: is there any constraint on $q_{max}$ ? there may exist trivial but useless upper bound for some loss functions.
   3. Suggest give more introductions about canonical SCM, to make the paper more self-contained.
3. about Colored MNIST experiment: I don't get how this experiments serves. Do you want to show the proposed bound is correct? However, there is no mention of the value of $R_{P*}(h)$ here.

**Limitations:**

see above.

---

> ### Author Rebuttal · Authors · 2024-08-06
>
> Thank you for your thoughtful review, we appreciate the positive feedback. In the following response we address comments and concerns pointed out by the reviewer. Please let us know if we can help clarify any part of it.
>
> > **Q 1.1.** *The paper organization and presentation seem complicated to convey the idea of the paper (see below). I believe there are existing works that assume some prior information regarding target domain and then derive bounds for the target loss in DG problems. If my understanding is correct, then I would like to treat the current work as an alternative way of encoding information about the target domain.*
>
> The current work (and the field of causal transportability as a whole) can be interpreted as a more general version of the transfer learning problem that operates using qualitative information about the commonalities and differences among the causal mechanisms of different domains (instances are [3,4,9,16,22] in the paper). As the reviewer suggests, other lines of research make different types of assumptions. One alternative is invariance learning methods, mentioned in the introduction, that exploit statistical invariances to induce robustness guarantees. We make sure to include more details in the related work section of the Appendix.
>
>
> > **Q 1.2.** *Then the current way of presentation and organization indeed make it complicated to understand the paper. In particular, I feel that examples 1 and 2 somehow distract reader's attention, while it is better to directly formulate the central problem of the paper.*
>
> Please notice that introducing Def. 4 earlier in the manuscript is challenging since it involves notions of transportability and partial identification. We believe that Examples 1 and 2 can be a useful for contextualizing the problem for the general reader and illustrate the definition of domain discrepancies, selection diagrams, etc. to set a common ground for discussing the challenge of the domain generalization problem.
>
> > **Q 1.3** *”It is not clear how this bound can in turn help improve general DG problems.”*
>
> Thank you for this question.
>
> Interpreting your question literally: “How does the bound returned by Neural-TR in Alg. 1 help improve the task of learning a predictor with robustness guarantees?” The bound improves general DG in the sense that Neural-TR is used explicitly by CRO in Alg. 2 that provably returns a predictor with optimal worst-case performance. In other words, finding the worst-case performance of predictors gives us a basis for comparing them, and eventually finding the one with the minimum worst-case risk, through CRO.
>
> Interpreting your question at a higher level: “How practical is CRO and its assumptions in real-world problems of domain generation?” We could start by noting that to solve any instance of the domain generalization problem, some notion of relevance between target domain and the source domains is required – as if domains are allowed to be arbitrarily different, transfer learning would be impossible. The advantage of CRO in practical problems is that it is completely non-parametric, making no assumptions on the distributional family of the variables involved. Moreover, selection diagrams only require the qualitative specification of commonalities and discrepancies across domains without needing to specify the underlying functional form of the domains.
>
> In case we didn’t fully grasp the intended meaning of “general DG”, could you please rephrase your question?
>
> > **Q 2.** *”Other questions: line 125: what does j indicate in the Sij notation? definition 4: is there any constraint on q_max? there may exist trivial but useless upper bound for some loss functions. Suggest give more introductions about canonical SCM, to make the paper more self-contained.”*
>
> $S_{ij}$ should read $S_i$, apologies for the typo. Def. 4 does not constrain $q_{max}$ a priori. Indeed, trivial upper bounds exist for all problems, e.g. fix $q_{max}$ be the upper end-point of the range of the loss function.  Note that $q_{max}$ obtained by the Neural-TR algorithm is guaranteed to be a tight bound in the limit, meaning that there exists no better (valid) upper bound, as shown in Thm. 2.
>
> > **Q 3.** *“about Colored MNIST experiment: I don't get how this experiments serves. Do you want to show the proposed bound is correct? However, there is no mention of the value of R_P* here.”*
>
> The motivation behind the Colored MNIST example is to compute the worst-case risk of different classifiers, as characterized by the source data and the selection diagram; these classifiers in our experiments are ERM and IRM, but the same analysis applies to any arbitrary classifier. The upper-bound for the risk (i.e., the worst-case risk) that is obtained with Neural-TR procedure, as suggested by Thm. 2, is asymptotically tight.
>
> In our analysis, we report in line 343 the value $R_{P^*}(h)$ of the classifiers $h:=\\{h_{ERM},h_{IRM}\\}$ on the worst-case $P^*$ found by Neural-TR. In addition, we supplement this analysis with an evaluation of CRO's classifier on all datasets. Those experiments are described and reported in the global rebuttal. In short, our experiments on evaluating CRO for simulated examples and on the Colored MNIST examples highlight the fact that CRO's output has the best worst-case risk among all classifiers, and its performance is contrasted to ERM and IRM.

---

> > ### Comment · Reviewer_S6HF · 2024-08-08
> >
> > Thanks for your reponse that mostly addresses my concerns.
> >
> > - regarding "general DG problem": thanks for pointing this out and I should not use "general" here. To be accurate, I mean these problems or datatsets that appear in many deep learning papers, e.g., ColoredMNIST, VLCS, etc.  (or check the paper: In Search of Lost Domain Generalization, ICLR 2021). How does the result of this paper improve emprical performance of these tasks?
> >
> > - regarding $q_max$ in Def. 4: if there is no further constraint, then there are many functions (e.g., every bounded function for some loss functions) that are partially transportable. Does this matter here?

---

> ### Author Response · Authors · 2024-08-09
>
> > "regarding "general DG problem": thanks for pointing this out and I should not use "general" here. To be accurate, I mean these problems or datatsets that appear in many deep learning papers, e.g., ColoredMNIST, VLCS, etc. (or check the paper: In Search of Lost Domain Generalization, ICLR 2021). How does the result of this paper improve emprical performance of these tasks?"
>
> Thank you for the clarification. We can give a concrete answer for the Colored MNIST dataset, where CRO entails a classifier with optimal worst-case generalization performance subject to variation in the correlation between the image color and label. The global rebuttal provides a description of that experiment (first bullet point and attached PDF), with figures describing the learning process of CRO as well as explicit comparisons with baseline algorithms.
> More generally, recall that CRO can be viewed as a meta-algorithm that is designed to exploit background information on the discrepancies and invariances between source and target domains, and guarantees optimal classifiers subject to these constraints. In this sense, if, for a particular DG task, background information can be specified (in the form of selection diagrams), CRO can be applied and will deliver a classifier that achieves best worst-case performance, as in the Colored MNIST experiment. Note, however, that background information might not necessarily be available for an arbitrary task defined in "In Search of Lost Domain Generalization, ICLR 2021". Out-of-the-box comparisons with more data-driven baselines, such as ERM and IRM, should be considered with care as these algorithms exploit different sets of assumptions that may not be appropriate in all settings.
>
> > "regarding qmax in Def. 4: if there is no further constraint, then there are many functions (e.g., every bounded function for some loss functions) that are partially transportable. Does this matter here?"
>
> Our objective is to infer the tightest upper bound qmax that satisfies the constraints (observational equivalence and structural invariance) following the definition of partial transportability. This notion is useful as it characterizes statements such as "the query is partially transportable with upper bound qmax" even though bounded functions are, by definition, already bounded from above. Considering the bounds in the context of worst-case performance of classifiers for the DG task, what you mention is correct since, with symmetric 0-1 loss, a trivial upper-bound for the risk of all classifiers is $R_{P^*}(h) \leq 1$, however, this bound is not informative or useful for classification in the DG problem. The bounds achieved by Neural-TR procedure are tightest valid bounds considering the source data and the domain knowledge, and provide a basis for comparing the DG performance of candidate classifiers.

---

> > ### Comment · Reviewer_S6HF · 2024-08-10
> >
> > Thanks for further clarifications. I main my score and am happy to see this paper accepted.

---

### Author Rebuttal · Authors · 2024-08-07

In this global rebuttal, we take the opportunity to discuss experimental results of CRO using the figures in the attached pdf as a support.

**Summary.** In the domain generalization task, the source domains $\mathcal{M}^1,\mathcal{M}^2,\dots,\mathcal{M}^K$ ,and the target domain $\mathcal{M}^*$ must be related for any learning to take place. The relatedness of the domains is expressed via a causal graph called selection diagram $\mathcal{G}^\Delta$ that encodes assumptions about the causal structure within each domain, as well as match/mismatch of mechanisms across the source and target domains. The source data $\mathbb{P} = \\{P^1,P^2,\dots,P^K\\}$ together with the selection diagram $\mathcal{G}^\Delta$ characterize the set of SCMs that are compatible as the target domain. Through canonical (Thm. 1) and neural (Thm. 2) parameterization, we implement procedures that obtain an upper-bound w.r.t. $\mathbb{P},\mathcal{G}^\Delta$ for an arbitrary target quantity; a especial case of our interest is the risk of a given classifier $h$ under the target distribution denoted by $R_{P^*}(h)$. This risk upper-bound is tight in the limit of data and expressivity of the parameterization, and represents the worst-case performance of the classifier at hand w.r.t. $\mathbb{P},\mathcal{G}^\Delta$. Neural-TR enhances the optimization procedure by cleverly decomposing the optimization objective, and computing the terms that are readily available from the source data to reduce parameter size and increase sample efficiency. Further, we introduce the CRO algorithm that uses Neural-TR as a subroutine and searches for a classifier with small worst-case risk. We prove that CRO terminates and outputs a classifier ideal for the domain generalization task which has the smallest worst-case risk across all classifiers. Below, we describe in detail how CRO operates.

**CRO.** We start with a random classifier; one might start with an ERM warm-start or any classifier of choice. At the first iteration, we use Neural-TR to compute the worst-case performance of the classifier at hand, witnessed by an NCM that entails the source data $\mathbb{P}$ and induces the selection diagram $\mathcal{G}^\Delta$. Next, we draw samples $D^{*1}$ from this NCM, and add them to a collection of datasets $\mathbb{D}$ that we maintain through the CRO procedure. Finally, we update the classifier at hand to be the minimizer of the maximum [empirical] risk over the datasets in $\mathbb{D}$. Since $\mathbb{D} = \\{D^{*1}\\}$ has only one dataset, the classifier updates to be the ERM of $D^{*1}$. We repeat the above process; at the second iteration, we use Neural-TR again to find yet another NCM that is compatible with  $\mathbb{P}, \mathcal{G}^\Delta$ while it incurs the worst risk on the classifier at hand. We draw samples $D^{*2}$ from this NCM, and add them to a collection of datasets $\mathbb{D}$. Finally, we update our classifier to be the minimizer of the maximum [empirical] risk over the datasets $\mathbb{D} =  \\{D^{*1}, D^{*2}\\}$. The process continues util the maximum risk witnessed by datasets in $\mathbb{D}$ converges to the true worst-case risk witnessed by Neural-TR.

In the attached pdf, you can find:

- **Figure 1: CRO on Colored-MNIST.** In section 5.2. we discussed Colored-MNIST, an instance of domain generalization task where the learner must classify the digits based on their colored image. The relationship between the color and the label is prone to change across the domains, and classifiers that rely on the color feature fail to generalize. In the two source domains, we intentionally set a high correlation between the color and the digit, thus classifiers that seek optimizing for risk in the source domains inevitably pick color as a determinant feature. We use this example to illustrate both the training process of CRO as well as the performance of the final classifier $h_{\mathrm{CRO}}$. We find that CRO converges in three iterations to an optimal predictor (in the worst-case sense) that ignores the color of the digit and instead makes a prediction based on the shape of the digit. CRO achieves an error of approximately 0.25, as shown in Figure 1d, which is theoretically optimal in this experiment. For comparison, the worst-case errors of ERM and IRM were evaluated to be 0.9 and 0.6 respectively. We also note that the poor performance of the baseline algorithms should not be blatantly compared to that of CRO, since CRO has access to background information that can not be communicated with the baseline algorithms. In this sense, CRO can be viewed as a meta-algorithm that operates with a broad range of assumptions encoded in a certain format (i.e., the selection diagram), while the baseline algorithms lack this capacity, and therefore, CRO is able to find the theoretically optimal classifier for domain generalization while the baseline algorithms fail to achieve that and perform poorly in the worst-case scenario.

- **Figure 2: CRO on Examples 2 & 3.** Examples 2 and 3 are synthetic instances of the domain generalization task that are small enough to be analyzed in fine granularity. In Example 2, three classifiers are considered (Table 1); total feature set, causal feature set, and non-causal feature set. Surprisingly enough, the non-causal feature set yields the best risk in the held-out domain. In Example 3, we considered a simple example with binary $X$ and $Y$. In this example, the considered classifiers are $0,1,X,\neg X$. Using Neural-TR, we compute the worst-case risk of considered classifiers in both examples, as reported in Figures 4a \& 4b in the original manuscript, and 2b \& 2d in the attached pdf. We also run CRO in both examples, and find that it discovers the best worst-case classifier in both examples; non-causal feature set in Example 2 and $\neg X$ in Example 3. Figures 2a \& 2c in the attached pdf demonstrate runs of Neural-TR for the classifier generated by CRO.

---

### Decision · Program_Chairs · 2024-09-25

**Decision:**

Accept (poster)

**Comment:**

This paper presents a novel and principled approach to addressing transportability problems within domain generalization. The theoretical results are sound and potentially useful. Given the positive feedback from all reviewers, I recommend this paper for acceptance.